Report

# Enzyme promiscuity shapes adaptation to novel growth substrates

Gabriela I Guzmán[1], Troy E Sandberg[1] [ID], Ryan A LaCroix[1], Ákos Nyerges[2] [ID], Henrietta Papp[3], Markus de Raad[4], Zachary A King[1], Ying Hefner[1], Trent R Northen[4], Richard A Notebaart[5], Csaba Pál[2], Bernhard O Palsson[1,6,7] [ID], Balázs Papp[2] & Adam M Feist[1,6,*] [ID]

## Abstract

Evidence suggests that novel enzyme functions evolved from low-level promiscuous activities in ancestral enzymes. Yet, the evolutionary dynamics and physiological mechanisms of how such side activities contribute to systems-level adaptations are not well characterized. Furthermore, it remains untested whether knowledge of an organism's promiscuous reaction set, or underground metabolism, can aid in forecasting the genetic basis of metabolic adaptations. Here, we employ a computational model of underground metabolism and laboratory evolution experiments to examine the role of enzyme promiscuity in the acquisition and optimization of growth on predicted non-native substrates in *Escherichia coli* K-12 MG1655. After as few as approximately 20 generations, evolved populations repeatedly acquired the capacity to grow on five predicted non-native substrates—D-lyxose, D-2-deoxyribose, D-arabinose, m-tartrate, and monomethyl succinate. Altered promiscuous activities were shown to be directly involved in establishing high-efficiency pathways. Structural mutations shifted enzyme substrate turnover rates toward the new substrate while retaining a preference for the primary substrate. Finally, genes underlying the phenotypic innovations were accurately predicted by genome-scale model simulations of metabolism with enzyme promiscuity.

**Keywords** adaptive evolution; enzyme promiscuity; genome-scale modeling; systems biology

**Subject Categories** Evolution; Genome-Scale & Integrative Biology; Metabolism

**Mol Syst Biol. (2019) 15: e8462**

## Introduction

Understanding how novel metabolic pathways arise during adaptation to environmental changes remains a central issue in evolutionary biology. The prevailing view is that enzymes often display promiscuous (i.e., side or secondary) activities and evolution takes advantage of such pre-existing weak activities to generate metabolic novelties (Jensen, 1976; Copley, 2000; Schmidt *et al*, 2003; Khersonsky & Tawfik, 2010; Huang *et al*, 2012; Nam *et al*, 2012; Näsvall *et al*, 2012; Voordeckers *et al*, 2012; Notebaart *et al*, 2014). However, it remains to be fully explored how these metabolic novelties are achieved via mutation events during periods of adaptation in short-term evolution experiments. Do genetic elements associated with promiscuous activities mutate mostly early on in adaptation when the initial innovative phenotype of growth on a new nutrient source is observed (Copley, 2000; Barrick & Lenski, 2013; Mortlock, 2013) or do promiscuous activities continue to play a role throughout the optimization process of continued fitness improvement on a non-native nutrient source (Barrick & Lenski, 2013)? In this work, mutational events that resulted in the ability of an organism to grow on a new, non-native carbon source were examined. These types of innovations have previously been linked to beneficial mutations that endow an organism with novel capabilities and expand into a new ecological niche (Wagner, 2011; Barrick & Lenski, 2013). Further, mutational events that were associated with more gradual enhancements of growth fitness (Barrick & Lenski, 2013) on the non-native carbon source were also examined. Such gradual improvements may stem from mutational events leading to regulatory improvements that fine-tune expression of desirable or undesirable pathways or possibly the fine-tuning of enzyme kinetics or substrate specificity of enzymes involved in key metabolic pathways (Copley, 2000; Barrick & Lenski, 2013). Enzyme promiscuity has been prominently linked to early mutation events, where mutations enhancing secondary activities may result in dramatic phenotypic improvements or new capabilities (Khersonsky & Tawfik, 2010;

1 Department of Bioengineering, University of California, San Diego, La Jolla, CA, USA
2 Synthetic and Systems Biology Unit, Institute of Biochemistry, Biological Research Centre of the Hungarian Academy of Sciences, Szeged, Hungary
3 Virological Research Group, Szentágothai Research Centre, University of Pécs, Pécs, Hungary
4 Environmental Genomics and Systems Biology Division, Lawrence Berkeley National Laboratory Berkeley, Berkeley, CA, USA
5 Laboratory of Food Microbiology, Wageningen University and Research, Wageningen, The Netherlands
6 Novo Nordisk Foundation Center for Biosustainability, Technical University of Denmark, Lyngby, Denmark
7 Department of Pediatrics, University of California, San Diego, La Jolla, CA, USA
*Corresponding author. Tel: +1 858 534 9592; E-mail: afeist@ucsd.edu

Barrick & Lenski, 2013). Therefore, in this work, we explored a diverse range of evolutionary routes taken during adaptation to new carbon sources. Specifically, we examined the role of enzyme promiscuity in both early mutations linked to innovative phenotypes and growth-optimizing mutations throughout various short-term laboratory evolution experiments.

A second open question in understanding the role of enzyme promiscuity in adaptation concerns our ability to predict the future evolution of broad genetic and phenotypic changes (Papp et al, 2011; Lässig et al, 2017). While there has been an increasing interest in studying empirical fitness landscapes to assess the predictability of evolutionary routes (de Visser & Krug, 2014; Notebaart et al, 2018), these approaches assess predictability only in retrospect. There is a need for computational frameworks that forecast the specific genes that accumulate mutations based on mechanistic knowledge of the evolving trait. A recent study suggested that a detailed knowledge of an organism's promiscuous reaction set (the so-called "underground metabolism"; D'Ari & Casadesús, 1998) enables the computational prediction of genes that confer new metabolic capabilities when artificially overexpressed (Notebaart et al, 2014). However, it remains unclear whether this approach could predict evolution in a population of cells adapting to a new nutrient environment through spontaneous mutations. First, phenotypes conferred by artificial overexpression might not be accessible through single mutations arising spontaneously. Second, and more fundamentally, mutations in distinct genes may lead to the same phenotype. Such alternative mutational trajectories may render genetic evolution largely unpredictable. Furthermore, computational approaches can aid in predicting and discovering overlapping physiological functions of enzymes (Guzmán et al, 2015; Notebaart et al, 2018), but these have also yet to be explored in the context of adaptation. In this study, we address these issues by performing controlled laboratory evolution experiments to adapt Escherichia coli to predicted novel carbon sources and by monitoring the temporal dynamics of adaptive mutations.

## Results

### Computational prediction and experimental evolution of non-native carbon source utilizations

Based on our knowledge of underground metabolism, we utilized a genome-scale model of E. coli metabolism that includes a comprehensive network reconstruction of underground metabolism (Notebaart et al, 2014) to test our ability to predict evolutionary adaptation to novel (non-native) carbon sources. This model was previously shown to correctly predict growth on non-native carbon sources if a given enabling gene was artificially overexpressed in a growth screen (Notebaart et al, 2014). This previous work identified a list of ten carbon sources that the native E. coli metabolic network is not able to utilize for growth in simulations but that can be utilized for growth in silico with the addition of a single underground reaction (Appendix Table S1). Based on this list—as well as substrate cost, availability, and solubility properties to maximize compatibility with our laboratory evolution procedures—we selected seven carbon sources (D-lyxose, D-tartrate, D-2-deoxyribose, D-arabinose, ethylene glycol, m-tartrate, monomethyl succinate) that cannot be utilized by wild-type E. coli MG1655 but are predicted to be growth-sustaining carbon sources after adaptive laboratory evolution.

Next, we initiated laboratory evolution experiments to adapt E. coli to these non-native carbon sources. Adaptive laboratory evolution experiments were conducted in two distinct phases: first, a "weaning/dynamic environment" (Copley, 2000; Mortlock, 2013) stage during which cells acquired the ability to grow solely on the non-native carbon sources and, second, a "static environment" (Barrick & Lenski, 2013) stage during which a strong selection pressure was placed to select for the fastest growing cells on the novel carbon sources (Fig 1A).

During the "weaning/dynamic environment" stage of laboratory evolution experiments (Fig 1A, see Materials and Methods), E. coli was successfully adapted to grow on five non-native substrates individually in separate experiments. Duplicate laboratory evolution experiments were conducted in batch growth conditions for each individual substrate and in parallel on an automated adaptive laboratory evolution (ALE) platform using a protocol that uniquely selected for adaptation to conditions where the ancestor (i.e., wild type) was unable to grow (Fig 1A; LaCroix et al, 2015). In the weaning phase, E. coli was dynamically weaned off of a growth-supporting nutrient (glycerol) onto the novel substrates individually (Fig 1A, Appendix Table S2). A description of the complex passage protocol is given in the Fig 1 legend and expanded in the methods for both phases of the evolution. This procedure successfully adapted E. coli to grow on five out of seven non-native substrates, specifically, D-lyxose, D-2-deoxyribose, D-arabinose, m-tartrate, and monomethyl succinate. Unsuccessful cases could be attributed to various experimental and biological factors such as experimental duration limitations, the requirement of multiple mutation events, or stepwise adaptation events, as observed in an experiment evolving E. coli to utilize ethylene glycol (Szappanos et al, 2016).

The "static environment" stage of the evolution experiments consisted of serially passing cultures in the early exponential phase of growth in order to select for cells with the highest growth rates (Fig 1A). Cultures were grown in a static media composition environment containing a single non-native carbon source. Marked and repeatable increases in growth rates on the non-native carbon sources were observed in as few as 180–420 generations (Appendix Table S1). Whole-genome sequencing of clones was performed at each distinct growth rate "jump" or plateau during the static environment phase (see arrows in Fig 1B, Appendix Fig S1). Such plateaus represent regions where a causal mutation has fixed in a population and it was assumed that the mutation(s) enabling the jump in growth rate were stable and maintained throughout the plateau region (LaCroix et al, 2015). Thus, clones were isolated at any point within this plateau region where frozen stock samples were available (LaCroix et al, 2015).

### Modeling with underground metabolism accurately predicted key genes mutated during laboratory evolution experiments

To analyze genotypic changes underlying the nutrient utilizations, clones were isolated and sequenced shortly after an innovative growth phenotype was achieved; mutations were identified (see Materials and Methods) and analyzed for their associated causality (Fig 1B, Appendix Fig S1, Dataset EV1). Strong signs of parallel evolution were observed at the level of mutated genes in the

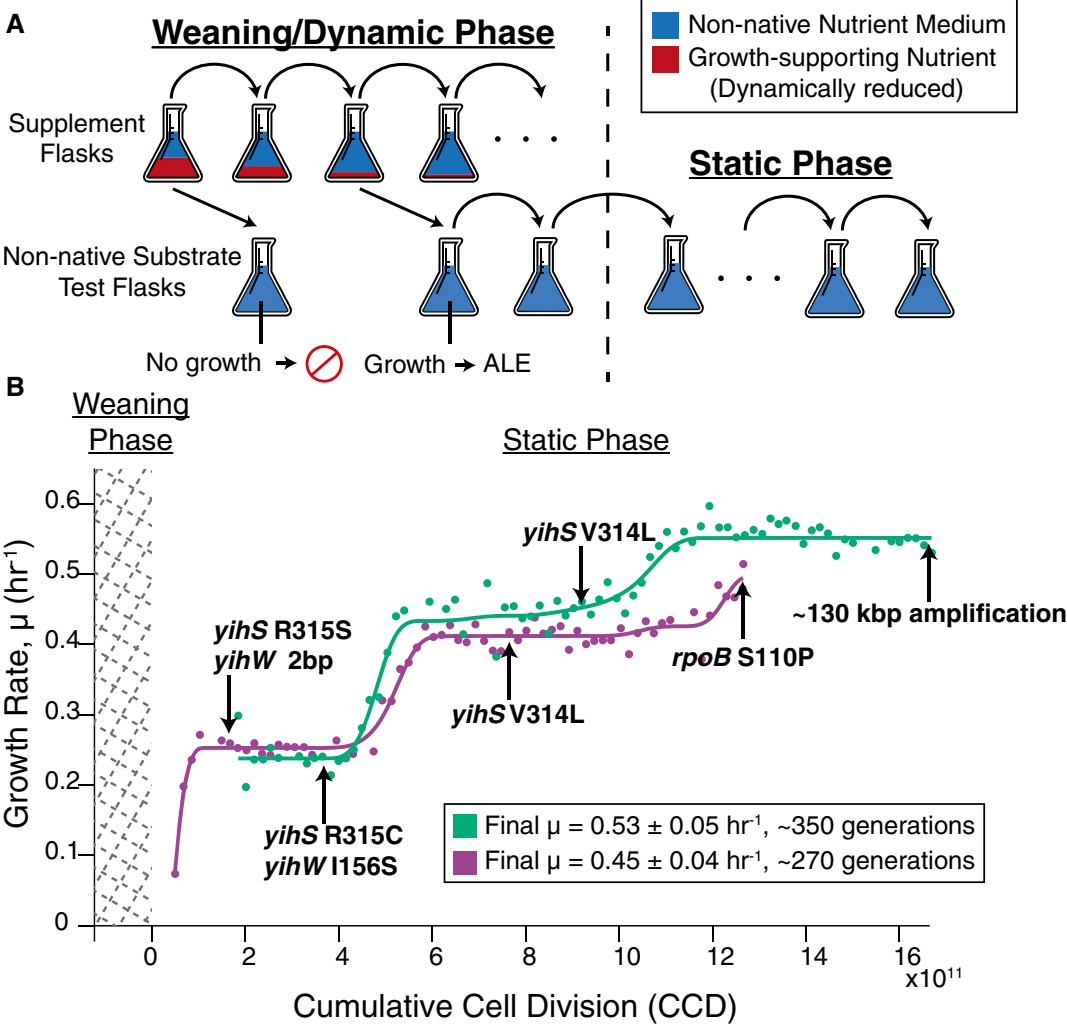

**Figure 1. Laboratory evolution method schematic and the growth trajectory of D-lyxose experiments.**

A  A schematic of the two-part adaptive laboratory evolution (ALE) experiments. The "weaning/dynamic environment" stage involved growing cells in supplemented flasks containing the non-native substrate (blue) and growth-promoting supplement (red). As cultures were serially passed, they were split into another supplemented flask as well as an "non-native substrate test flask" containing only the non-native nutrient (no supplement) to test for the desired evolved growth phenotype. The "static environment" stage consisted of selecting for the fastest growing cells and passing in mid log phase.

B  Growth rate trajectories for duplicate experiments (*n* = 2 evolution experiments per substrate condition) (green and purple) for the example case of D-lyxose. Population growth rates are plotted against cumulative cell divisions. Clones were isolated for whole-genome sequencing at notable growth rate plateaus as indicated by the arrows. Mutations gained at each plateau are highlighted beside the arrows (mutations arising earlier along the trajectory persisted in later sequenced clones).

replicate evolution experiments (Fig 1B, Appendix Fig S1, Table 1, Dataset EV1). Such parallelism provided evidence of the beneficial nature of the observed mutations and is a prerequisite for predicting the genetic basis of adaptation (Bailey *et al*, 2015). Mutations detected in the evolved isolated clones for each experiment demonstrated a striking agreement with such predicted "underground" utilization pathways (Notebaart *et al*, 2014). Specifically, for four out of the five different substrate conditions, key mutations were linked to the predicted enzyme with promiscuous activity, which would be highly unlikely by chance ($P < 10^{-8}$, Fisher's exact test; Table 1, Appendix Fig S2). Not only were the specific genes (or their direct regulatory elements) mutated in four out of five cases, but few additional mutations (0–2 per strain, Dataset EV1) were observed

directly following the weaning phase, indicating that the innovative phenotypes observed required a small number of mutational steps and the method utilized was highly selective. For the one case where the prediction and observed mutations did not align—D-arabinose—a detailed inspection of the literature revealed existing evidence that three *fuc* operon-associated enzymes can metabolize D-arabinose—FucI, FucK, and FucA (LeBlanc & Mortlock, 1971). The mutations observed in the D-arabinose evolution experiments after the weaning stage were in the *fucR* gene (Table 1), a DNA-transcriptional activator associated with regulating the expression of the transcription units *fucAO* and *fucPIK* (Podolny *et al*, 1999). Thus, it was inferred that the strains evolved to grow on D-arabinose in our experiments were utilizing the *fuc* operon-associated enzymes to metabolize

**Table 1.** Key mutations associated with growth phenotypes after weaning phase

| Gene mutated | Substrate | Gene prediction | Protein change(s) (Experiment #) | Perceived impact (Structural (S) or Regulatory (R)) |
|---|---|---|---|---|
| yihS | D-Lyxose | yihS | R315S (1) | Substrate binding[a] (S) |
| | | | R315C (2) | Substrate binding[a] (S) |
| yihW | D-Lyxose | yihS | Frameshift (1) | Loss of function, large truncation (R) |
| | | | I156S (2) | - (R) |
| rbsK | D-2-Deox. | rbsK | N20Y (1) | - (S) |
| rbsR | D-2-Deox. | rbsK | Insertion Sequence (1) | Loss of function; increased rbsK expression (R) |
| 181 kbp and 281 kbp Regions | D-2-Deox. | rbsK | - (1) | Increased gene expression (R) |
| fucR | D-Arabinose | rbsK | D82Y (1) | Pfam: DeoRC C terminal substrate sensor domain[b] (R) |
| | | | S75R (1 and 2) | Pfam: DeoRC C terminal substrate sensor domain[b] (R) |
| | | | *244C (2) | - (R) |
| dmlA | m-Tartrate | dmlA | A242T (1) | - (S) |
| dmlR/dmlA | m-Tartrate | dmlA | Intergenic −50/−53 (2) | Sigma 70 binding: close proximity to −10 of dmlRp3 promoter[c] (R) |
| | | | Intergenic −35/−68 (2) | dmlRp3 promoter region[c] (R) |
| ybfF/seqA | Mon. Succ. | ybfF | Intergenic −73/−112 (1) | Sigma 24 binding: −35 of ybfFp1 promoter[c] (R) |
| | | | Intergenic −51/−123 (2) | Sigma 24 binding: −10 of ybfFp1 promoter[c] (R) |

Substrates D-2-deoxyribose and monomethyl succinate are abbreviated D-2-Deox. and Mon. Succ., respectively. The detailed locations of the mutations listed in this table are available in Dataset EV1 and Appendix Fig S3.
[a]Substrate binding information about YihS previously published (Itoh et al, 2008).
[b]Protein family information listed in the Pfam database (Finn et al, 2016).
[c]Promoter/sigma factor binding regions found on EcoCyc (Keseler et al, 2013) based on computational predictions (Huerta & Collado-Vides, 2003).

D-arabinose in agreement with prior work (LeBlanc & Mortlock, 1971). In this case, the genome-scale model did not identify the promiscuous reactions responsible for growth on D-arabinose because the promiscuous (underground) reaction database was incomplete (see section "Mutations in regulatory elements linked to increased expression of underground activities: D-arabinose evolution" for more details on D-arabinose metabolism).

In general, key mutations observed shortly after strains achieved reproducible growth on the non-native substrate could be categorized as regulatory (R) or structural (S) (Table 1). Of the fifteen mutation events outlined in Table 1, eleven were categorized as regulatory (observed in all five successful substrate conditions) and four were categorized as structural (three of five successful substrate conditions). For D-lyxose, D-2-deoxyribose, and m-tartrate evolution experiments, mutations were observed within the coding regions of the predicted genes, namely yihS, rbsK, and dmlA (Table 1, Appendix Fig S1). Regulatory mutations occurring in transcriptional regulators or within intergenic regions—likely affecting sigma factor binding and transcription of the predicted gene target—were observed for D-lyxose, D-2-deoxyribose, m-tartrate, and monomethyl succinate (Table 1). Observing more regulatory mutations is broadly consistent with previous reports (Mortlock, 2013; Toll-Riera et al, 2016). The regulatory mutations were believed to increase the expression of the target enzyme, thereby increasing the dose of the typically low-level side activity (Guzmán et al, 2015). This observation is consistent with "gene sharing" models of promiscuity and adaptation where diverging mutations that alter enzyme specificity are not necessary to acquire the growth innovation (Piatigorsky et al, 1988; Guzmán et al, 2015). Furthermore, although enzyme dosage could also be increased through duplication of genomic segments,

this scenario was not commonly observed shortly after the weaning phase of our experiments. The one exception was observed in the D-2-deoxyribose evolution experiment where two large duplication events (containing 165 genes (yqiG-yhcE) and 262 genes (yhiS-rbsK), respectively) were observed (Appendix Fig S3). Notably, one of these regions did include the rbsK gene with the underground activity predicted to support growth on D-2-deoxyribose (Table 1).

To identify the causal mutation events relevant to the observed innovative nutrient utilization phenotypes, each key mutation (Table 1) was introduced into the ancestral wild-type strain using the genome engineering method pORTMAGE (Nyerges et al, 2016). This genome editing approach was performed to screen for mutation causality (Herring et al, 2006) on all novel substrate conditions, except for monomethyl succinate, which only contained a single mutation (Table 1). Individual mutants were isolated after pORTMAGE reconstruction, and their growth was monitored in a binary fashion on the growth medium containing the non-native substrate over the course of 1 week. These growth tests revealed that single mutations were sufficient for growth on D-lyxose, D-arabinose, and m-tartrate (Appendix Table S3). Interestingly, in the case of D-2-deoxyribose, an individual mutation (either the RbsK N20Y or the rbsR insertion mutation) was not sufficient for growth, thereby suggesting that the mechanism of adaptation to this substrate was more complex. To address this, a pORTMAGE library containing the RbsK N20Y and rbsR insertion mutations individually and in combination was grown on three M9 minimal medium + 2 g l⁻¹ D-2-deoxyribose agar plates alongside a wild-type MG1655 ancestral strain control. The large duplications in the D-2-deoxyribose strain (Table 1) could not be reconstructed due to the limitations of the pORTMAGE method. After 10 days of incubation, visible colonies

could be seen resulting from the reverse engineered library, but not from the wild-type strain (Appendix Fig S4A). Subsequently, 16 colonies were chosen and colony PCR was performed to sequence the regions of rbsK and *rbsR* where the mutations were introduced (Appendix Fig S4B). All 16 colonies sequenced contained both the RbsK N20Y and *rbsR* insertion mutations. Fifteen of the 16 colonies showed an additional mutation at RbsK residue Asn14—7 colonies showed a AAT to GAT codon change resulting in an RbsK N14D mutation and 8 colonies showed a AAT to AGT codon change resulting in an RbsK N14S mutation. The Asn14 residue has been previously associated with ribose substrate binding of the ribokinase RbsK enzyme (Sigrell *et al*, 1999). Only one of the 16 colonies sequenced did not acquire the residue 14 mutation, but instead acquired a GCA to ACA codon change at residue Ala4 resulting in an RbsK A4T mutation. It is unclear if the additional mutations occurred spontaneously during growth prior to plating, but it is possible that these Asn14 and Ala4 residue mutations were introduced at a low frequency during MAGE-oligonucleotide DNA synthesis (< 0.1% error rate at each nucleotide position) (Isaacs *et al*, 2011; Nyerges *et al*, 2018). In either case, these results suggested that the observed mutations in rbsK and rbsR enabled growth on the non-native D-2-deoxyribose substrate and that there was a strong selection pressure on the ribokinase underground activity. Further, there were multiple ways to impact *rbsK,* as both duplication events and structural mutations (Table 1) or multiple structural mutations were separately observed in strains which grew solely on D-2-deoxyribose. Overall, these causality assessments support the notion that underground activities can open short adaptive paths toward novel phenotypes and may play prominent roles in innovation events.

**Examination of growth-optimizing evolutionary routes**

Once the causality of the observed mutations was established, adaptive mechanisms required for further optimizing or fine-tuning growth on the novel carbon sources were explored. Discovery of these growth-optimizing activities was driven by a systems-level analysis consisting of mutation, enzyme activity, and transcriptome analyses coupled with computational modeling of optimized growth states on the novel carbon sources. Out of the total set of 41 mutations identified in the static phase of the evolution experiments

(Datasets EV1 and EV2), a subset (Table 2) was explored. This subset consisted of genes that were repeatedly mutated in replicate experiments or across all endpoint sequencing data on a given non-native carbon source. To unveil the potential mechanisms for improving growth on the non-native substrates, the transcriptome of initial and endpoint populations (i.e., right after the end of the weaning phase, and at the end of the static environment phase, respectively) was analyzed using RNA-seq. Differentially expressed genes were compared to genes containing optimizing mutations (or their direct targets) and targeted gene deletion studies were performed. Additionally, for the D-lyxose experiments, enzyme activity was analyzed to determine the effect of a structural mutation acquired in a key enzyme during growth optimization on the single non-native carbon source. Analysis of mutations in the static growth-optimizing phase led to identification of additional promiscuous enzyme activities above and beyond those causal mutation mechanisms identified shortly after the weaning phase. Enzyme promiscuity appeared to play a role in the adaptive routes utilized to optimize growth in at least three of the five nutrient conditions (Table 2). Detailed analyses of these results are described in the following sections in case studies for the D-lyxose, D-arabinose, and D-2-deoxyribose evolution experiments.

**Structural mutations linked to shifting substrate affinities: D-lyxose evolution**

A clear example of mutations involved with optimization was those acquired during the D-lyxose experiments that were linked to enhancing the secondary activity of YihS (Tables 1 and 2). Structural mutations were hypothesized to improve the enzyme side activity to support the growth optimization state, and this effect was experimentally verified. The effects of structural mutations on enzyme activity were examined for the YihS isomerase enzyme that was mutated during the D-lyxose evolution (Fig 1B, Table 1). The activities of the wild-type YihS and three mutant YihS enzymes (YihS R315S, YihS V314L + R315C, and YihS V314L + R315S) were tested *in vitro*. A cell-free *in vitro* transcription and translation system (Shimizu *et al*, 2001; de Raad *et al*, 2017) was used to express the enzymes and examine conversions of D-mannose to D-fructose (a primary activity; Itoh *et al*, 2008) and D-lyxose to D-xylulose (side

**Table 2. Mutations associated with growth optimization during static phase**

| Gene mutated | Substrate | Mutation type | Proposed impact | Associated with underground activity? |
|---|---|---|---|---|
| *yihS* | D-Lyxose | V314L SNP | Improved D-Lyxose affinity | Yes |
| 131 kbp Region | D-Lyxose | Large Duplication (129 genes) | Increased *xylB* expression | No |
| *rbsB, rbsB/rbsK* | D-2-Deoxyribose | 902 bp Deletion spanning gene and intergenic region | Increased *rbsK* expression | Yes |
| 183 kbp Region | D-2-Deoxyribose | Large Deletion (171 genes) | Decreased expression unnecessary genes | Maybe |
| *araC* | D-Arabinose | 6 bp Deletion, SNP | Increased *araB* expression | Yes |
| *ygbI* | m-Tartrate | 20 bp Deletion, SNP | Increased *ygbJKLMN* expression | Maybe |
| *pyrE* | D-Lyxose[a], m-Tartrate | Duplication[a], Intergenic | Increased *pyrE* expression | No |

The detailed locations of the mutations listed in this table are available in Dataset EV1, Appendix Fig S8, and Appendix Fig S10.
[a]*pyrE* is located in the large region of duplication (second entry of table).

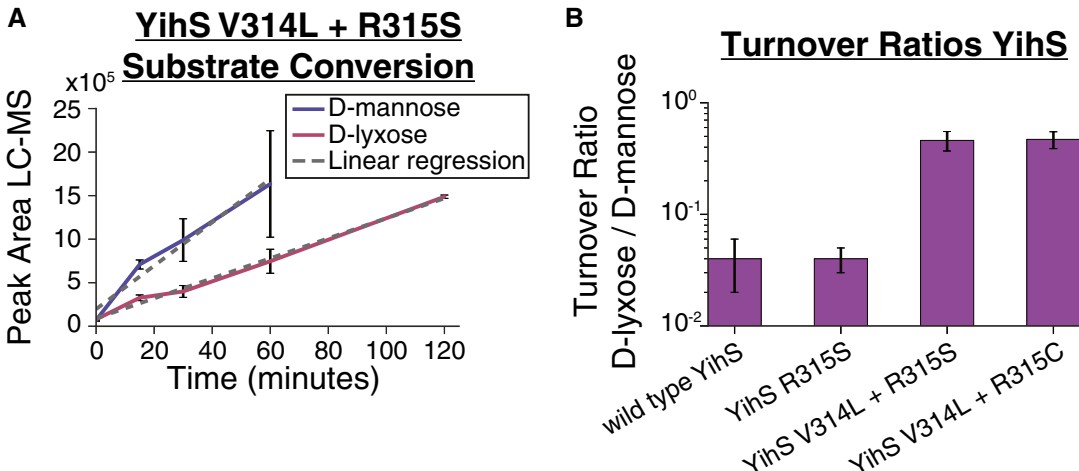

**Figure 2. Evaluation of enzymatic activity for the wild-type and mutated promiscuous enzyme, YihS.**

A  YihS V314L + R315S mutant enzyme activity on D-mannose and D-lyxose. LC-MS was used to analyze YihS activity at saturating substrate concentrations to compare turnover rates on each substrate. Product formation was followed over time at a constant enzyme concentration. Turnover rates were calculated using linear regression ($n = 3$ replicates for each enzyme, Dataset EV4). The error bars represent standard deviation ($n = 3$) of the peak area.

B  Turnover ratios of substrate conversion of D-lyxose/D-mannose are shown for the wild-type YihS and mutant YihS enzymes. A ratio < 1 indicates a higher turnover rate on D-mannose compared to D-lyxose. Error bars represent standard error ($n = 3$) calculated from the linear regression analysis.

activity) (Fig 2A, Appendix Fig S5). The ratios of the turnover rates of D-lyxose to the turnover rates of D-mannose were calculated and compared (Fig 2B). Although the single-mutant YihS enzyme did not show a significant change compared to wild type, the double-mutant YihS enzymes showed approximately a 10-fold increase in turnover ratio of D-lyxose to D-mannose compared to wild type ($P < 0.0003$, ANCOVA). These results suggest that the mutations shifted the affinity toward the innovative substrate (enzyme side activity), while still retaining an overall preference for the primary substrate, D-mannose (ratio < 1). This is in agreement with "weak trade-off" theories of the evolvability of promiscuous functions (Khersonsky & Tawfik, 2010) in that only a small number of mutations are sufficient to significantly improve the promiscuous activity of an enzyme without greatly affecting the primary activity.

### Mutations in regulatory elements linked to increased expression of underground activities: D-arabinose evolution

An important growth rate optimizing mutation was found in the D-arabinose experiments and occurred as a result of an *araC* gene mutation, a DNA-binding transcriptional regulator that regulates the *araBAD* operon involving genes associated with L-arabinose metabolism (Bustos & Schleif, 1993). Based on structural analysis of AraC (Fig 3A), the mutations observed in the two independent parallel experiments likely affect substrate binding regions given their proximity to a bound L-arabinose molecule (RCSB Protein Data Bank entry 2ARC; Soisson *et al*, 1997), possibly increasing its affinity for D-arabinose. Expression analysis revealed that the *araBAD* transcription unit associated with AraC regulation (Gama-Castro *et al*, 2016) was the most highly upregulated set of genes (expression fold increase ranging from approximately 45–65× for Exp 1 and 140–200× for Exp 2, $q < 10^{-4}$, FDR-adjusted *P*-value) in both experiments (Fig 3B). Further examination of these upregulated genes revealed that the ribulokinase (AraB) has a similar kcat on four

2-ketopentoses (D/L-ribulose and D/L-xylulose) (Lee *et al*, 2001) despite the fact that *araB* is consistently annotated to only act on L-ribulose (EcoCyc) (Keseler *et al*, 2013) or L-ribulose and D/L-xylulose (BiGG Models; King *et al*, 2016). It was thus reasoned that AraB was catalyzing the conversion of D-ribulose to D-ribulose 5-phosphate in an alternate pathway for metabolizing D-arabinose (Fig 3C) and this was further explored.

The role of the AraB pathway in optimizing growth on D-arabinose was analyzed both computationally and experimentally. Parsimonious flux balance analysis (pFBA; Feist & Palsson, 2010; Lewis *et al*, 2010) simulations demonstrated that cell growth with AraB had a higher overall metabolic yield than growth with FucK (in simulations where only one of the two pathways was active, Appendix Fig S6). This supported the hypothesis that mutants with active AraB can achieve higher growth rates than those in which it is not expressed. This simulation result signaled the possibility of a growth advantage for using the AraB pathway and thus was explored experimentally. Experimental growth rate measurements of clones carrying either an *fucK* knockout or *araBAD* gene knock-outs showed that the FucK enzyme activity was essential for growth on D-arabinose for all strains analyzed (strains isolated after initial growth on the single non-native carbon source and strains isolated at the end of the static environment phase) (Fig 3D, Appendix Table S4). However, removal of *araB* from endpoint strains reduced the growth rate to the approximate growth rate of the initially adapted strain (Fig 3D). This finding suggested that the proposed AraB pathway (Fig 3C) was responsible for enhancing the growth rate and therefore qualified as fitness optimization.

Putting these computational and experimental results in the context of previous work, a similar pathway has been described in mutant *Klebsiella aerogens* W70 strains (St Martin & Mortlock, 1977). It was suggested that the D-ribulose-5-phosphate pathway (i.e., the AraB pathway) is more efficient for metabolizing D-arabinose than the D-ribulose-1-phosphate pathway (i.e., the FucK

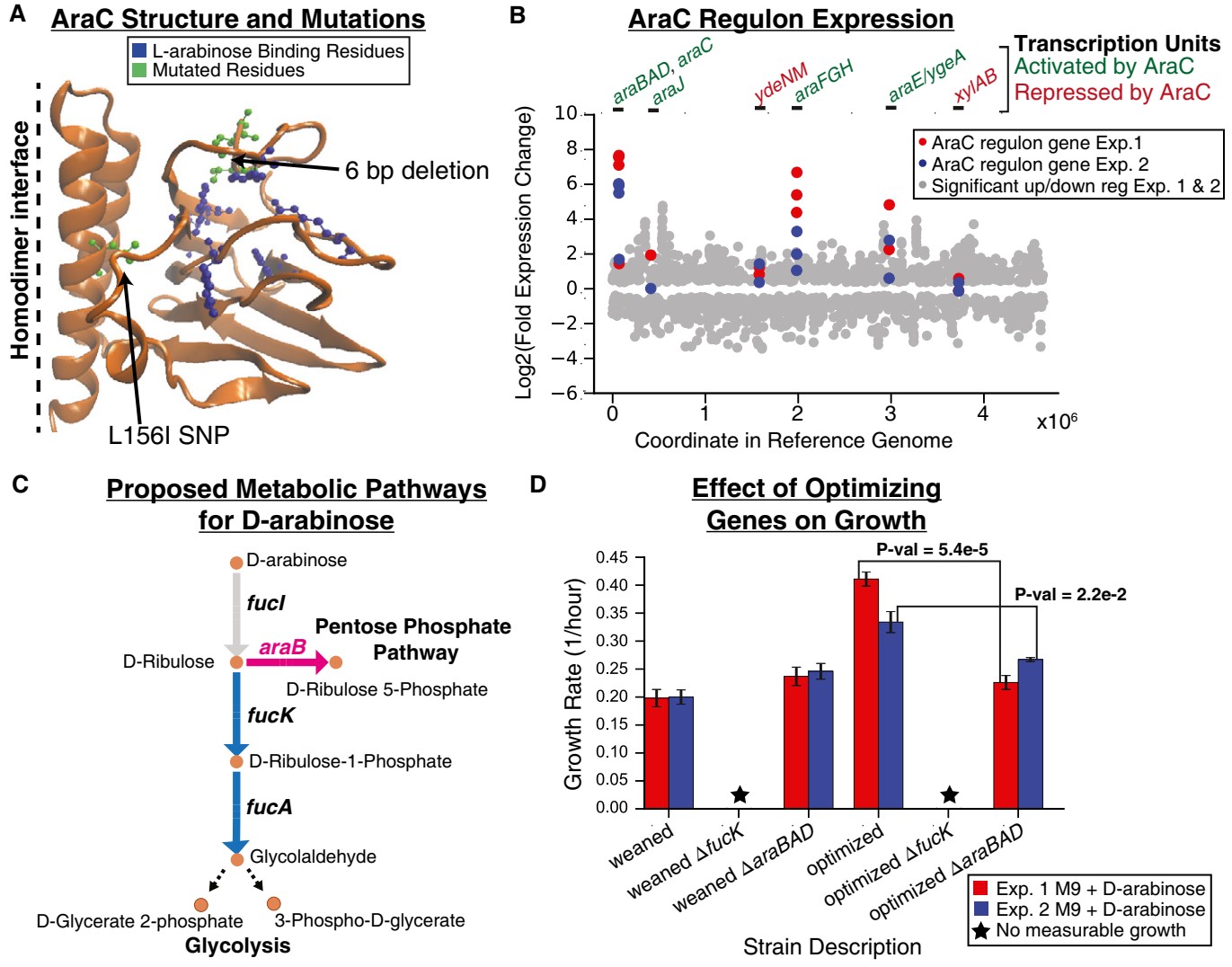

**Figure 3. Optimization mutation analysis for D-arabinose evolution experiments.**

A Structural mutations observed in sequencing data of Experiments (Exp.) 1 and 2 (green) as well as residues previously identified as important for binding L-arabinose (blue) are highlighted on one chain of the AraC homodimer protein structure. The six base pair deletion observed in Exp. 1 appears to be most clearly linked to affecting substrate binding.

B Expression data (RNA-seq) for significantly differentially expressed genes (*q*-value < 0.05, FDR-adjusted *P*-value, *n* = 2 biological replicates for each condition). Scatter plot shows log2(fold change) of gene expression data comparing endpoint to initial populations for Exp. 1 and Exp. 2 (gray dots) with the location of the gene in the reference genome as the *x*-axis. Those genes that are associated with AraC transcription units are highlighted (red dots for Exp. 1 and blue dots for Exp. 2). Above the plot, the transcription units are labeled green if AraC activates expression (in the presence of arabinose) or red if AraC represses expression of those genes.

C The proposed two pathways for metabolizing D-arabinose. The pink pathway is enabled by the optimizing mutations observed in *araC*.

D Growth rate analysis of various weaned (starting point of static phase) and optimized (endpoint of static phase) strains with or without *fucK* or *araB* genes knocked out. Strains were grown in triplicate (*n* = 3) on M9 minimal media with D-arabinose as the sole carbon source. The colored bars represent the calculated mean growth rate, and the error bars represent the standard deviation. The *P*-values reported were calculated using a two-sided Welch's *t*-test.

pathway), because the FucK pathway requires that three enzymes (FucI, FucK, and FucA) recognize secondary substrates (St Martin & Mortlock, 1977). The conclusion of St Martin and Mortlock supports the role of the mutations observed here in *araC*. In summary, enzymatic side activities of both the *fuc* operon (innovative mutations) and *ara* operon (optimizing mutations) encoded enzymes were important for the adaptation to efficiently metabolize D-arabinose.

Computational and expression analyses suggested that a similar mechanism of amplification of growth-enhancing promiscuous

activities played a role in the m-tartrate optimization regime. Similar to the D-arabinose experiments, both independent evolutions on m-tartrate possessed a mutation in the predicted transcription factor, *ygbI*. This mutation was associated with the overexpression of a set of genes (*ygbI, ygbJ, ygbK, ygbL, ygbM,* and *ygbN*) with likely promiscuous activity (Appendix Supplementary Text and Appendix Fig S7). Further experiments, however, are required to better elucidate the mechanism and involvement of *ygb* operon-associated enzymes in the metabolism of m-tartrate.

**Genome-scale modeling suggests a role of segmental genome duplication and deletion in adaptation: D-2-deoxyribose and D-lyxose evolutions**

Large genome duplications and deletions were observed in the D-lyxose and D-2-deoxyribose evolution experiments. These events were examined using a genome-scale metabolic model to understand their potential impact on strain fitness. First, we considered whether the large deletion event in the D-2-deoxyribose evolution Exp. 1 (Table 2, Appendix Fig S8) contained genes involved in metabolism. The 171 deleted genes were compared to those genes included in the genome-scale model of metabolism used in this study. It was found that 44 metabolic genes were located in this region of deletion (Dataset EV3). Flux variability analysis (FVA; Mahadevan & Schilling, 2003) simulations revealed that none of the 44 genes are individually necessary for optimal growth under these conditions (Dataset EV3). In fact, all 44 genes can be deleted at once from the genome-scale model without affecting the simulated growth rate. It was also interesting to note that 18 of the 44 genes were highly expressed in the initially evolved population after weaning and thus significantly down-regulated ($\log_2$(fold change) $< -1$, $q$-value $<$ 0.025) by the large deletion event in the endpoint evolution population (Dataset EV3 and Appendix Fig S9). These observations are in agreement with previously reported findings that cells acquire mutations that reduce the expression of genes not required for growth during evolution and thus allow the cell to redirect resources from production of unnecessary proteins to increasing growth functions (Utrilla *et al*, 2016). Furthermore, there was an additional mutation observed at the same time as the large deletion, namely a smaller 902 bp deletion spanning a major part of the rbsB gene and into the intergenic region upstream of rbsK (Table 2, Dataset EV1, and Dataset EV2). The perceived impact of this deletion was to further increase the expression of rbsK, the gene associated with the underground activity required for growth on D-2-deoxyribose. The concept of removing enzymatic activities, and potentially multiple simultaneously, to increase fitness is an interesting avenue which, in this case, necessitates a significant number of additional experiments to confirm given the multiple genes affected.

While the YihS structural mutations appeared to be the primary mutations responsible for optimizing growth on D-lyxose (Fig 2), a genome duplication event observed in Exp. 2 could play a role in improving the growth rate (Fig 1B, Table 2). The genome duplication event spanned a 131 kilobase pair region (Appendix Fig S10A) resulting in significant up-regulation of 76 genes (Appendix Fig S10B). Included in this gene set were *pyrE* and *xylB*, two genes identified by modeling as important for metabolizing D-lyxose. The first gene, *pyrE,* could enhance growth by increasing nucleotide biosynthesis (Conrad *et al*, 2009), and this gene is important for achieving optimal growth in genome-scale model simulations (Appendix Fig S10C). The *pyrE* gene might have also played a role in improving growth fitness in the m-tartrate evolution experiments where intergenic mutations upstream of the *pyrE* gene were observed in both replicate evolving endpoint populations (Table 2, Appendix Supplementary Text and Fig S7C, and Dataset EV2). Another gene in the large duplication event was *xylB*, encoding a xylulokinase, which might be catalyzing the second step in the metabolism of D-lyxose (Appendix Fig S10D). Simulating increased flux through the xylulokinase reaction in an approach similar to a phenotypic phase plane analysis (Ibarra *et al*, 2003) improved the growth rate on D-lyxose (Appendix Fig S10C). Thus, increased expression of *xylB* and *pyrE* as a result of the duplication event in the Exp. 2 endpoint strains could be important for enhancing growth on the non-native substrate D-lyxose. While follow-up experiments over-expressing these genes individually are necessary to conclusively establish the causal role of increased *pyrE* and *xylB* expression, this study provides a high-level picture of the complex mechanisms at work in adaptation to new carbon sources, from structural and regulatory mutations to large-scale deletions and duplications.

## Discussion

The results of this combined computational analysis and laboratory evolution study show that enzyme promiscuity can play a major role in an organism's adaptation to novel growth environments. It was demonstrated that enzyme side activities can confer a fitness benefit and open routes for achieving innovative growth states. Further, it was observed that mutation events that enabled growth on non-native carbon sources could be structural or regulatory in nature and that in four out of the five substrate conditions examined, a single innovative mutation event related to a promiscuous activity was sufficient to support growth. Strikingly, it was demonstrated that network analysis of underground activities could be used to predict these evolutionary outcomes. Furthermore, beyond providing an evolutionary path for innovation, it was demonstrated that enzyme promiscuity aided in the optimization of growth in multiple, distinct ways. It was shown that structural mutations in an enzyme with a secondary activity with a selective advantage could improve the substrate affinity for the non-native carbon source as was observed in the D-lyxose evolution experiments. Finally, it was observed that enzyme promiscuity beyond the enzyme activity initially selected for could open secondary novel metabolic pathways to more efficiently metabolize the new carbon source. This was most clearly observed in the D-arabinose evolutions in which *fuc* operon-associated enzyme activities were required for the initial innovative phenotype of growth, and then, the *ara* operon activities were associated with further growth optimization.

While this study showcases the prominent role of enzyme promiscuity in evolutionary adaptations, there is room for follow-up work to strengthen the claims and broaden implications. One strength of this study was examining multiple short-term laboratory evolution experiment conditions (i.e., multiple non-native substrates) in duplicate; however, the number of non-native substrates explored was still on a relatively small scale and the results were a collection of case studies. Next steps could include broadening the number of non-native substrates as well as conducting laboratory evolution experiments with many more replicates and over longer periods of time. Furthermore, there were many mutations, particularly acquired during the static environment phase of experiments, that were not thoroughly examined for causality. This is evident in the case of the small and large deletion in the D-2-deoxyribose evolution experiment found in the clone isolated after the final fitness jump. With hundreds of genes removed from the genome, a deep dive into this event is necessary to unravel the impact, and modeling along with transcriptomics was suggested as a tool to aid in this process. Finally, further studies could examine

the trade-offs of enhancing secondary enzyme activities while maintaining a primary activity. This was touched upon while examining the influence of mutations on YihS enzyme activities; however, a more thorough look at enzyme kinetics for multiple cases (such as those observed in DmlA and RbsK (Table 1)) could provide a clearer picture of mutation trade-offs.

The results of this study are relevant to our understanding of the role of promiscuous enzymatic activities in evolution and for utilizing computational models to predict the trajectory and outcome of molecular evolution (Papp *et al*, 2011; Lässig *et al*, 2017). Here, we demonstrated that genome-scale metabolic models that include the repertoire of enzyme side activities can be used to predict the genetic basis of adaptation to novel carbon sources. As such, genome-scale models and systems-level analyses are likely to contribute significantly toward representing the complex implications of promiscuity in theoretical models of molecular evolution (Lässig *et al*, 2017).

# Materials and Methods

## Genome-scale model simulations

The *i*JO1366 (Orth *et al*, 2011; model accessible for download at: http://bigg.ucsd.edu/models/iJO1366) version of the genome-scale model of *Escherichia coli* K-12 MG1655 was utilized in this study as the wild-type model before adding underground reactions related to five carbon substrates (D-lyxose, D-2-deoxyribose, D-arabinose, m-tartrate, monomethyl succinate) as previously reported (Notebaart *et al*, 2014). The underground reactions previously reported were added to *i*JO1366 using the constraint-based modeling package COBRApy (Ebrahim *et al*, 2013). The version of the *i*JO1366 model with the added underground reactions explored in this study is provided in Model EV1. All growth simulations used parsimonious flux balance analysis (pFBA) (Lewis *et al*, 2010). Growth simulations were performed by maximizing flux through the default biomass objective function (a representation of essential biomass compounds in stoichiometric amounts) (Feist & Palsson, 2010). To simulate aerobic growth on a given substrate, the exchange reaction lower bound for that substrate was adjusted to $-10$ mmol $gDW^{-1}$ $h^{-1}$. Predictions of positive growth phenotypes have been demonstrated to be robust against the exact value of the uptake rate given that they are in a physiological range (Edwards & Palsson, 2000). We note that the metabolic network without the underground reactions is completely incapable of providing growth on any of the carbon sources examined and as such, the predictions can be considered qualitative predictions that are only dependent on the network structure.

For the pFBA results shown in Appendix Fig S6, Appendix Fig S7C, and Appendix Fig S10C, the effect of changing flux through a reaction of interest on growth rate was examined by sampling through a range of flux values (changing the upper and lower flux bounds of the reaction) and then optimizing the biomass objective function. This resulted in a set of flux values and growth rate pairs that were then plotted in the provided figures. Flux variability analysis (FVA) simulations (Mahadevan & Schilling, 2003) were implemented in COBRApy (Ebrahim *et al*, 2013) with a growth rate cutoff of 99% of the maximum biomass flux. FVA was used to analyze the potential growth impact of the large deletion event from the

D-2-deoxyribose evolution. For the D-2-deoxyribose simulations, a glyceraldehyde demand reaction (Orth & Palsson, 2012) was added to prevent a false-negative gene knockout result with the removal of *aldA*, as previously described. Additionally, *aldA* isozyme activity for the reaction ALDD2x was also added to the model for D-2-deoxyribose simulations. This isozyme addition was based on literature findings (Rodríguez-Zavala *et al*, 2006).

## Laboratory evolution experiments

The bacterial strain utilized in this study as the starting strain for all evolutions and MAGE manipulations was an *E. coli* K-12 MG1655 (ATCC 4706). Laboratory evolution experiments were conducted on an automated platform using a liquid handling robot as previously described (Sandberg *et al*, 2014; LaCroix *et al*, 2015). As described above, the experiments were conducted in two phases, a "weaning/dynamic environment" phase and an "static environment" phase. At the start of the weaning phase, cultures were serially passaged after reaching stationary phase in a supplemented flask containing the non-native carbon source at a concentration of 2 g $l^{-1}$ and the growth-supporting supplement (glycerol) at a concentration of 0.2%. Cultures were passaged in stationary phase and split into another supplemented flask and a test flask containing only the non-native carbon source at a concentration of 2 g $l^{-1}$. As the weaning phase progressed, the concentration of the growth-supporting nutrient was adjusted to maintain a target max OD600 (optical density 600 nm) of 0.5 as measured on a Tecan Sunrise plate reader with 100 μl of sample. This ensured that glycerol was always the growth limiting nutrient. If growth was not observed in the test flask within 3 days, the culture was discarded; however, once growth was observed in the test flask, this culture was serially passaged to another test flask. Once growth was maintained for three test flasks, the second phase of the evolution experiments commenced—the static environment phase. The static environment phase was conducted as in previous studies (Sandberg *et al*, 2014; LaCroix *et al*, 2015). The culture was serially passaged during mid-exponential phase so as to select for the fastest growing cells on the innovative carbon source. Growth was monitored for a given flask by taking OD600 measurements at four time points, targeted to span an OD600 range of 0.05–0.3, with sampling time based on the most recently measured growth rate and the starting OD. Samples were also periodically taken and stored in 25% glycerol stocks at $-80$°C for reference and for later sequencing analysis. The evolution experiments were concluded once increases in the growth rate were no longer observed for several passages.

Growth data from the evolution experiments were analyzed with an in-house MATLAB package. Growth rates were calculated for each flask during the "static environment" phase of the evolution experiments by taking the slope of a least-squares linear regression fit to the logarithm of the OD measurements vs. time. Calculated growth rates were rejected if fewer than three OD measurements were sampled, the range of OD measurements was < 0.2 or > 0.4, or if the $R^2$ correlation for the linear regression was < 0.98. Generations of growth for each flask were calculated by taking log([flask final OD]/[flask initial OD])/log(2), and the cumulative number of cell divisions (CCD) was calculated based on these generations as described previously (Lee *et al*, 2011). Growth rate trajectory curves (Fig 1B, Appendix Fig S1) were produced in MATLAB by fitting a

monotonically increasing piecewise cubic spline to the data as reported previously (Sandberg *et al*, 2014; LaCroix *et al*, 2015).

## Growth media composition

All strains were grown in M9 minimal medium. The M9 minimal medium was composed of the carbon source at a concentration of 2 g l$^{-1}$ unless otherwise specified (e.g., during the "weaning/ dynamic" phase of the ALE experiments the total amount of carbon source varied as the growth-supporting nutrient concentration was dynamically decreased). Carbon sources were purchased from Sigma-Aldrich (D-(-)-Lyxose 99% catalog #220477, 2-Deoxy-D-Ribose 97% catalog #121649, D-(-)-Arabinose ≥ 98% catalog #A3131, meso-Tartaric acid monohydrate ≥ 97% catalog #95350, and mono-Methyl hydrogen succinate 95% catalog #M81101). The growth-supporting nutrient used was glycerol. Other components of the M9 minimal medium were 0.1 mM CaCl$_2$, 2.0 mM MgSO$_4$, 6.8 g l$^{-1}$ Na$_2$HPO$_4$, 3.0 g l$^{-1}$ KH$_2$PO$_4$, 0.5 g l$^{-1}$ NaCl, 1.0 g l$^{-1}$ NH$_4$Cl, and trace elements solution. A 4,000× trace element solution consisted of 27 g l$^{-1}$ FeCl$_3$* 6 H$_2$O, 2 g l$^{-1}$ NaMoO$_4$* 2 H$_2$O, 1 g l$^{-1}$ CaCl$_2$* H$_2$O, 1.3 g l$^{-1}$ CuCl$_2$* 6 H$_2$O, 0.5 g l$^{-1}$ H$_3$BO$_3$, and concentrated HCl dissolved in double-distilled H$_2$O and sterile filtered. The final concentration in the media of the trace elements solution was 1×.

## Whole-genome sequencing and mutation analysis

Colonies were isolated and selected on Lysogeny Broth (LB) agar plates and grown in M9 minimal media + the corresponding non-native carbon source prior to genomic DNA isolation. For population sequencing conducted for endpoint strains (Dataset EV2), samples were taken directly from glycerol frozen stocks and grown in M9 minimal media + the corresponding non-native carbon source prior to genomic DNA isolation. Genomic DNA was isolated using the Macherey-Nagel Nucleospin Tissue Kit using the support protocol for bacteria provided by the manufacturer user manual. The quality of genomic DNA isolated was assessed using Nanodrop UV absorbance ratios. DNA was quantified using Qubit dsDNA high-sensitivity assay. Paired-end whole-genome DNA sequencing libraries were generated utilizing either a Nextera XT kit (Illumina) or KAPA HyperPlus kit (Kapa Biosystems). DNA sequencing libraries were run on an Illumina Miseq platform with a paired-end 600 cycle v3 kit.

DNA sequencing fastq files were processed utilizing the computational pipeline tool, *breseq* (Deatherage & Barrick, 2014) version 0.30.0 with bowtie2 (Langmead & Salzberg, 2012) version 2.2.6, aligning reads to the *E. coli* K-12 MG1655 genome (NC000913.3; Datasets EV1 and EV2). For the clone and population samples sequenced in this study, the average of percent mapped reads was > 90%, the average mean coverage was 106 reads, the average total reads was 2.08E6 reads, and the average read length was 271. When running the breseq tool, the input parameters for clonal samples were options -j 8, and the input parameters for population samples were options -p -j 8—polymorphism-frequency-cutoff 0.0. For further information regarding breseq mutation call/read alignment methods, please refer to the breseq methods publication (Deatherage & Barrick, 2014) and documentation. Additionally, identification of large regions of genome amplification was identified using a custom python script that utilizes aligned files to identify regions with more than 2× (minus standard deviation) of mean read depth coverage. DNA-seq mutation datasets are also available on the public database ALEdb 1.0.2 (http://aledb.org; Phaneuf *et al*, 2019).

## Enzyme activity characterization

All enzymes used in this study were generated by cell-free *in vitro* transcription and translation using the PURExpress *in vitro* Protein Synthesis Kit (New England Biolabs). Linear DNA templates utilized in all cell-free *in vitro* transcription and translation reactions were generated by PCR from dsDNA blocks encoding the enzymes with transcription and translations elements synthesized by Integrated DNA Technologies. Linear DNA templates were purified and concentrated using phenol/chloroform extraction and ethanol precipitation. The encoded enzymes were produced using PURExpress according to manufacturer's protocol with linear DNA templates concentrations of 25 ng/1 μl reaction.

The activities of the wild-type YihS and three mutant YihS enzymes toward D-Mannose and D-Lyxose over time were determined using LC/MS. Substrate (10 mM) was added to 7.5 μl of PURExpress reaction in a buffered solution (50 mM Tris, 100 mM KCl, 10 mM MgCl$_2$, pH 8) for a total volume of 250 μl and incubated at 37°C. At different time points (0, 15, 30, 60, 120, 240, and 1,320 min), 10 μl of samples was taken and quenched with 90 μl of LC/MS grade ethanol. Next, samples were dried under vacuum (Savant SpeedVac Plus SC110A) and resuspended in 50 μl of LC/MS grade methanol/water (50/50 v/v). The samples were filtered through 0.22-μm microcentrifugal filtration devices and transferred to 384-well plate for LC/MS analysis. An Agilent 1290 LC system equipped with a SeQuant® ZIC®-HILIC column (100 mm × 2.1 mm, 3.5 μm 200 Å, EMD Millipore) was used for separation with the following LC conditions: solvent A, H$_2$O with 5 mM ammonium acetate; solvent B, 19:1 acetonitrile:H$_2$O with 5 mM ammonium acetate; timetable: 0 min at 100% B, 1.5 min at 100% B, 6 min at 65% B, 8 min at 0% B, 11 min at 0% B, 12.5 min at 100% B, and 15.5 min at 100% B; 0.25 ml min$^{-1}$; column compartment temperature of 40°C. Mass spectrometry analyses were performed using an Agilent 6550 quadrupole time-of-flight mass spectrometer. Agilent software Mass Hunter Qualitative Analysis (Santa Clara, CA) was used for naïve peak finding and data alignment. Analysis of covariance (ANCOVA) was used to determine whether the slopes of mutants for both xylose and mannose are significantly different from the wild-type slopes. Detailed instrument information and data are provided in Appendix Table S6 and Dataset EV4.

## pORTMAGE Library Construction/Isolation of individual mutants

Mutations were introduced and their corresponding combinations accumulated during the laboratory evolution experiments into the ancestral *E. coli* strain using pORTMAGE recombineering technology (Nyerges *et al*, 2016). ssDNA oligonucleotides, carrying the mutation or mutations of interest, were designed using MODEST (Bonde *et al*, 2014) for *E. coli* K-12 MG1655 (ATCC 4706). To isolate individual mutants, a single pORTMAGE cycle was performed separately with each of the 15 oligos in *E. coli* K-12 MG1655 (ATCC 4706) + pORTMAGE3 (Addgene ID: 72678) according to a previously described pORTMAGE protocol (Nyerges *et al*, 2016). Following transformation, cells were allowed to recover

overnight at 30°C and were plated to Luria Bertani (LB) agar plates to form single colonies. Presence of each mutation or mutation combinations was verified by High-Resolution Melting (HRM) colony PCRs with Luminaris HRM Master Mix (Thermo Scientific) in a Bio-Rad CFX96 qPCR machine according to the manufacturer's guidelines. Mutations were confirmed by capillary-sequencing. pORTMAGE oligonucleotides, HRM PCR, and sequencing primers are listed in Dataset EV5.

### D-2-deoxyribose pORTMAGE library agar plate growth experiments

The pORTMAGE library containing *rbsR* and *rbsK* mutations separately and in combination was used to conduct growth experiments on M9 minimal medium + 2 g l$^{-1}$ D-2-deoxyribose agar plates. The pORTMAGE library frozen glycerol stock composed of the library grown on LB medium, as well as the wild-type *E. coli* MG1655 frozen glycerol stock, also an LB grown stock, was used to inoculate M9 minimal medium + 2 g l$^{-1}$ D-2-deoxyribose or M9 minimal medium + 2 g l$^{-1}$ glycerol and grown at 37°C overnight. The overnight cultures, which contained some residual LB medium and glycerol from the frozen stock, underwent several generations each and were visibly dense (OD600 = ~0.5–1.0). The next day, 1 ml of the overnight cultures was pelleted by centrifugation at 5,000 *g* for 5 min. After pelleting, cells were washed and resuspended in 1 ml of M9 minimal medium + no carbon source. Pelleting and washing was repeated two more times to remove any residual glycerol carbon source or LB media components, and the final resuspension was used for plating. Both the pORTMAGE library and wild-type cells (either from the glycerol or D-2-deoxyribose pre-culture, as specified in Appendix Fig S4A) were plated using a 10-µl inoculation loop on either half of three M9-minimal medium + 2 g l$^{-1}$ D-2-deoxyribose agar plates (Appendix Fig S4A). The agar plates were made by mixing a 2× solution of D-2-deoxyribose M9 minimal medium with a 2× autoclaved solution of agar (18 g agar in 0.5 l of Milli-Q water). The plates were incubated at 37°C for a total of 9–10 days.

After 9–10 days of incubation, 16 pORTMAGE library colonies were picked from the three D-2-deoxyribose plates for colony PCR and sequencing (Appendix Fig S4A). Colony PCR was conducted (Qiagen HotStarTaq Master Mix Kit) with the primer sequences listed in Appendix Table S5 for rbsK and rbsR. DNA sequencing of PCR products was conducted by Eton Bioscience Inc using their SeqRegular services. The sequencing results are summarized in the main text and in Appendix Fig S4B. Sequencing alignments were conducted using the multiple sequence alignment tool Clustal Omega (Sievers *et al*, 2011; Appendix Fig S4B).

### RbsK comparison to DeoK/kinases in other Enterobacteriaceae

Protein sequence alignment was conducted for the *E. coli* MG1655 RbsK N20Y mutant sequence from this study and DeoK sequences reported for *E. coli* strains (Bernier-Febreau *et al*, 2004; Monk *et al*, 2013), three pathogenic (AL862, 55989, and CFT073) and one commensal (EC185), as well as the DeoK sequence reported for *S. enterica* serovar Typhi (Tourneux *et al*, 2000). The sequence alignments were performed using the multiple sequence alignment package, T-Coffee (Notredame *et al*, 2000) (Appendix Supplementary Text and Fig S11).

### Individual mutant growth test

Isolated mutants were tested for growth over the course of 1 week (Appendix Table S3). Individual colonies were isolated on LB agar plates and used to inoculate pre-cultures grown overnight in 2 ml of glucose M9 minimal liquid media in 10-ml tubes. The following morning, pre-cultures were pelleted at 2,000 *g* and gently resuspended (by pipetting) in M9 minimal medium without a carbon source and this spinning and resuspension was repeated twice to wash the cells of residual glucose. The final resuspension was in 2 ml of M9 minimal medium without a carbon source. The growth test tubes consisting of 2 ml of M9 minimal medium plus the corresponding innovative carbon source were inoculated with the washed cells at a dilution factor of 1:200. Growth was monitored over the course of 1 week by visually inspecting for increased cellular density, noting that the cultures had become opaque from cell growth. Once growth was observed, colony PCR was conducted (Qiagen HotStarTaq Master Mix Kit) with the primer sequences listed in Appendix Table S5. DNA sequencing of PCR products was conducted by Eton Bioscience Inc using their SeqRegular services. DNA sequencing was utilized to confirm the designed mutations were as expected and to confirm that no other mutations had been acquired in the regions of interest during the growth test.

### RNA sequencing

RNA sequencing data were generated under conditions of aerobic, exponential growth on M9 minimal medium plus the corresponding non-native carbon source (D-lyxose, D-2-deoxyribose, D-arabinose, or m-tartrate). Cells were harvested using the Qiagen RNA-protect bacteria reagent according to the manufacturer's specifications. Prior to RNA extraction, pelleted cells were stored at −80°C. Cell pellets were thawed and incubated with lysozyme, SuperaseIn, protease K, and 20% sodium dodecyl sulfate for 20 min at 37°C. Total RNA was isolated and purified using Qiagen's RNeasy minikit column according to the manufacturer's specifications. Ribosomal RNA (rRNA) was removed utilizing Ribo-Zero rRNA removal kit (Epicentre) for Gram-negative bacteria. The KAPA Stranded RNA-seq kit (Kapa Biosystems) was used for generation of paired-end, strand-specific RNA sequencing libraries. RNA sequencing libraries were then run on an Illumina HiSeq 2500 using the "rapid-run mode" with 2 × 35 paired-end reads.

Reads were mapped to the *E. coli* K-12 genome (NC_000913.2) using bowtie (Langmead *et al*, 2009). Cufflinks (Trapnell *et al*, 2010) was utilized to calculate the expression level of each gene in units per kilobase per million fragments mapped (FPKM). This information was then utilized to run cuffdiff (Trapnell *et al*, 2013) to calculate gene expression fold change between endpoint and initial growth populations (*n* = 2 biological replicates for each condition tested) using a geometric normalization and setting a maximum false discovery rate of 0.05. Gene expression fold change was considered significant if the calculated *q*-value (FDR-adjusted *P*-value of the test statistic) was smaller than 0.025 and after conducting a Benjamini-Hochberg correction for multiple-testing (values obtained from cuffdiff analysis). The RNA-seq data are available in the Gene Expression Omnibus (GEO) database under the accession number GSE114358.

**Metabolic map generation and data superimposition**

All metabolic pathway maps generated in Fig 3 and Appendix Fig S7, Appendix Fig S9, and Appendix Fig S10 were generated using the pathway visualization tool Escher (King *et al*, 2015).

**Bioscreen growth test of mutants**

Individual sequenced clones (Dataset EV1) from the D-arabinose evolution experiments (Exp. 1 and Exp. 2) along with the wild-type *E. coli* K-12 MG1655 strain were utilized for bioscreen growth tests and gene knockout manipulations. A P1-phage transduction mutagenesis protocol based on a previously reported method (Donath *et al*, 2011) was followed to replace the *fucK* gene in the evolution and wild-type strains with a Kanamycin resistance cassette from the *fucK* Keio strain (Baba *et al*, 2006). The BW25113 Keio collection strain is effectively missing the *araBAD* genes, so the *yabI* Keio strain was utilized for the P1-phage transduction of all strains to transfer this neighboring *araBAD* deletion along with the *yabI*-replaced Kanamycin resistance cassette. It was deemed that a *yabI* deletion would not significantly affect the results of the growth experiments since *yabI* is a non-essential inner membrane protein that is a member of the DedA family (Doerrler *et al*, 2013). *Escherichia coli* K-12 contains seven other DedA proteins, and it is only collectively that they are essential (Boughner & Doerrler, 2012).

The growth screens were conducted in a Bioscreen-C system machine. Pre-cultures were started from frozen stocks of previously isolated clones and grown overnight in M9 minimal medium + 0.2% glycerol. These pre-cultures were used to inoculate the triplicate bioscreen culture wells at 1:100 dilution of M9 minimal medium supplemented with either 2 g/l D-arabinose or 0.2% glycerol. The final volume for each well was 200 μl. The growth screen was conducted under continuous shaking conditions at 37°C. OD600 (optical density at 600 nm) readings were taken every 30 min over the course of 48 h. Growth rates were calculated using the tool Croissance (Schöning, 2017). The mean growth rates and standard deviation for each condition ($n = 3$) were calculated and reported in Appendix Table S4 and Fig 3D. The *P*-values reported in Fig 3D were calculated using a two-sided Welch's *t*-test.

# Data availability

The datasets and model produced in this study are available in the following databases:

- RNA-seq data: Gene Expression Omnibus GSE114358 (https://www.ncbi.nlm.nih.gov/geo/query/acc.cgi?acc=GSE114358)
- DNA-Seq mutation data: ALEdb 1.0.2 (http://aledb.org) as well as provided in Datasets EV1 and EV2.
- The genome-scale metabolic model is provided as Model EV1.

**Expanded View** for this article is available online.

## Acknowledgements
We would like to thank Richard Szubin for assistance with strain resequencing, as well as Elizabeth Brunk, Balázs Szappanos, and Bálint Kintses for helpful discussion. We also thank Suzanne Kosina for her assistance with LC-MS experiments. We also thank Samira Dahesh and the Nizet Lab for their assistance with Bioscreen experiments. This work was partially supported by The Novo Nordisk Foundation Grant Number NNF10CC1016517. TRN and MDR acknowledge funding from ENIGMA—Ecosystems and Networks Integrated with Genes and Molecular Assemblies (http://enigma.lbl.gov), a Scientific Focus Area Program at Lawrence Berkeley National Laboratory is based upon work supported by the U.S. Department of Energy, Office of Science, Office of Biological & Environmental Research under contract number DE-AC02-05CH11231. AN was supported by a PhD fellowship from the Boehringer Ingelheim Fonds. CP and BP acknowledge funding from the "Lendület" Programme of the Hungarian Academy of Sciences, The Wellcome Trust and GINOP-2.3.2-15-2016-00014 (EVOMER). CP was also supported by the European Research Council (H2020-ERC-2014-CoG) and BP by the National Research, Development and Innovation Office, Hungary (NKFIH grant KH125616).

## Author contributions
GIG, TES, RAL, TRN, RAN, CP, BP, BOP, and AMF designed the research; GIG, TES, RAL, YH, ZAK, AN, HP, and MR performed research; GIG, TES, ZAK, AN, MR, BP, and AMF analyzed data; and GIG, BP, and AMF wrote the paper.

## Conflict of interest
The authors declare that they have no conflict of interest.

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
