## [Review Process File · Molecular Systems Biology]

Enzyme promiscuity shapes adaptation to novel growth substrates

Gabriela I. Guzmán, Troy E. Sandberg, Ryan A. LaCroix, Ákos Nyerges, Henrietta Papp, Markus de Raad, Zachary A. King, Ying Hefner, Trent R. Northen, Richard A. Notebaart, Csaba Pál, Bernhard O. Palsson, Balázs Papp and Adam M. Feist.

Review timeline:

Submission date:	22 nd May 2018
Editorial Decision:	26 th June 2018
Revision received:	8 th November 2018
Editorial Decision:	11 th January 2019
Revision received:	8 th March 2019
Accepted:	14 th March 2019

Editor: Maria Polychronidou

Transaction Report:

1st Editorial Decision

26th June 2018

Thank you again for submitting your work to Molecular Systems Biology. We have now heard back from the three referees who agreed to evaluate your study. As you will see below, the reviewers think that the presented findings seem potentially interesting. They raise however a series of concerns, which we would ask you to address in a major revision.

Without repeating all the points listed below, one of the more fundamental issues raised by reviewers #1 and #2 refers to the distinction between the innovation and optimization phase of adaptive evolution. The reviewers think that the division into two phases does not seem justified and recommend revising the related conclusions. Moreover, they refer to the need to include additional experimental analyses to better support the main conclusions.

All other issues raised by the reviewers would need to be convincingly addressed. As you may already know, our editorial policy allows in principle a single round of major revision so it is essential to provide responses to the reviewers' comments that are as complete as possible. Please feel free to contact me in case you would like to discuss in further detail any of the issues raised by the reviewers.

REFeree REPORTS.

Reviewer #1:

Guzman et al combines experimentation and computation to understand: a) how the capacity to utilize novel energy/carbon sources using promiscuous enzymes evolves b) how well solutions can be predicted based on flux balance metabolic models. The paper's contribution to the first issues is encapsulated in the statements: "we demonstrate that enzyme promiscuity can be linked to fitness benefits in both the innovation and optimization stages of adaptive evolution." The contribution to

the second issue is summarized by: "genes underlying the phenotypic innovations were accurately predicted by genome-scale model simulations of metabolism with enzyme promiscuity." The authors nicely integrate experimental and computational techniques, present an extensive body of work, and experiments and analysis, with a few exceptions, were well performed. If revised, the paper is likely to be well received by sections of the experimental evolution and metabolic modelling communities; I am not entirely convinced about the size of these sections nor that it will attract a broader readership.

Specific concerns:

1. The authors emphasize the conclusion: "we demonstrate that enzyme promiscuity can be linked to fitness benefits in both the innovation and optimization stages of adaptive evolution". They present this as a major finding. I am not sure that this finding represents a substantial advancement of evolutionary biology. Why would enzyme promiscuity be beneficial in only one of these stages? Is that a theory-driven proposition? What is the rationale? Indeed, the authors do not further discuss the significance of this finding in the conclusions, suggesting that they are not convinced that this is a significant finding either. Second, it is not clear to me why optimizing mutations would not emerge in what in the design is called the innovation phase. Third, given that the allele frequencies are not tracked over time and clone dynamics are unknown, it seems dangerous to make assumptions about when mutations first emerge (pre-experiment, innovation and optimization phase), in what order, and to what degree their fitness benefits manifest in these phases. Fourth, is there really a clear qualitative functional distinction between "innovative" and "optimizing" mutations here? Can the authors reject that sequencing of mutations is not only a matter of chance, local mutation rates, or small differences in selection coefficients? The authors classify the mutation *rsbK N20Y* as an innovative mutation in Table 1, yet when reconstructed it does not seem to be sufficient for growth even after a week, according to Table S3. It strikes me as surprising since the innovative mutations were discovered initially by the innovation test lasting for only 3 days. The authors comment on this finding arguing that it's a complex scenario requiring additional regulatory mutations, but isn't that very close to the definition of an optimizing mutation? Why is not the reconstruction of this regulatory mutation done and tested? According to Figure S1, it appears that the first sampling for sequencing where the *rsbK* mutations are found is performed after substantial adaptation. Without phenotyping both *rsbK* mutations how do the authors reject the possibility that *rsbK N20Y* is an "optimizing" mutation in this case? I think this is a striking example of the seemingly arbitrary boundary between "innovative" and "optimizing" mutations from a biological perspective. In summary - I am not yet convinced by the clear separation the authors want to make between "innovation" and "optimization" and find it artificial and in danger of being near entirely subjective.

2. In the conclusions, the authors' introduce two alternative key conclusions: "First, side activities contributed to the establishment of novel metabolic routes that enabled or improved the utilization of a new nutrient source. Second, suppression of an undesirable underground activity that diverted flux from a newly established pathway conferred a fitness benefit." The first statement is well supported by the data but not entirely novel in itself. More problematically, I don't feel that they have sufficiently well supported the second statement. The authors postulate that a large deletion of 171 genes present at unknown frequencies in endpoint populations is the driver of adaptation at previous time points. It is unclear what the empirical support driving this conclusion is as there is no recurrence, no reconstructions and no time course sequencing. How do the authors reject the null hypothesis of the mutation being neutral? Further, the authors conclude that *AldA* deletion is likely the driver, because in the founder *Ald6* had the most highly expressed metabolic transcript under these conditions. There are multiple problems with this inference: the tenuous link between transcripts and protein activity, between high protein activity and deleterious effects of the protein, the assumption that the driver mutation must be metabolic rather than regulatory, the assumption that founder cell state reflects the cell state after innovation, and the assumption that none of the 171 deleted genes affect fitness individually or in combination (it seems rather unlikely that these would all be neutral). Finally, the authors support their statement with an FBA, stating that conversion of acetaldehyde to acetate has negative effects on growth rate. How large was the effect? What is the precision and accuracy of the model? What other fluxes have negative effects on growth rates in the model? Larger negative effects on growth rates? Are any of these genes in the deleted segment? What is the chance probability that any one of the genes associated with negative effects on the growth rates in the model would occur in a 171 gene deletion? Are there additional enzymes catalyzing the reaction in *E. coli*? *aldA* is claimed to be most significantly differentially expressed among the metabolic genes, but are there any other metabolic genes among the differentially

expressed genes? How does the modeling look for those genes compared to aldA? I cannot help feel that the evidence presented in 2.4. is quite circumstantial - the conclusion is not well supported by data, as stands. Several things could and should have been done, with deleting AldA from an innovated, not optimized clone and inserting AldA into an optimized clone being the most important.

3. I am surprised and a little concerned about the data presented in Table S3. The authors very nicely reconstruct many of the candidate innovative mutations in a founder background - excellent and laborious job. But they do not do a proper quantification of the effect of the mutation on growth rates in the growth conditions of interest. Instead they do a qualitative, subjective and presumably quite error prone assay "time to visual first growth". This rounded to "full days". This is not entirely convincing. I would like to see the growth rates and I would like to see the growth rates converted into selection coefficients (the authors equate growth rates with fitness so this should be a straight-forward transformation). It's unclear how a week was selected as the duration for the experiments underlying Table S3. As understood from SI material and methods the classification and discovery of an "innovative" mutation is visible growth after three days. It is quite remarkable that none of the mutants tested for Table S3 grows to visible growth in <3 days. Does this mean that none of the mutations tested for Table S3 actually were "innovative" mutations? Ideally, I would like to see some kind of perspective on whether these mutations realistically could explain the type of adaptation kinetics observed. In this context, I am a little concerned by statements such as (in the abstract). "After as few as 20 generations, the evolving populations repeatedly acquired the capacity to grow on predicted non-native substrates". First, the statement is ambiguous - it is unclear from where it is taken, how the number is calculated and what the authors really mean, explicitly? At a casual reading, 20 generations seems to imply very fast adaptation. Can the reconstructed mutations, which seem very weak, really explain that? It seems highly unlikely to me, especially assuming that the population starts out without standing variation. According to Table S2, 20 generations is nowhere to be seen. Something is amiss here.

4. The paper feels like a somewhat hastily composed and slightly disorganized product. Too many, somewhat disconnected follow-up threads joined together with an unclear rationale and sequencing and division into paragraphs of very uneven length. Typos, sentences with unclear syntax and grammar and duplicated refs that should have been weeded out before submission. Figure legends that in many cases lack key information on error bars, type and number of replicates and tests behind reported p-values. Mutations that often are annotated only as the affected gene. References to the wrong display item, or statements not supported by references to display items. It feels like a not sufficiently well crafted final product and the authors should perhaps go back to the drawing board.

5. The conclusion "genes underlying the phenotypic innovations were accurately predicted by genome-scale model simulations of metabolism with enzyme promiscuity" is probably well supported. However, the formal statistics underlying this conclusion is unclear (F1 score?) and the authors do not explain the erroneous prediction for D-arabinose. While the model missed the fuc operon associated genes, it is not clear why it did the prediction it did and why the predicted gene was not empirically validated. In Fig S2, which is supporting the main statement about how well the authors modeling findings can be experimentally observed, the genes fucI, fucK, fucA are claimed to be experimentally observed. I would expect to find these genes in either Fig S1 or in Table 1. I do not. Since these genes make up 100% of one of the cells in the contingency table it's surprising that I cannot find references to any of these genes in the rest of the paper in terms of results.

6. The M&M section and supporting figs is too concise and in some cases do not allow for a good evaluation of results. This includes a) explanations for choices, assumptions, and parameters estimates in the model and the robustness of conclusions to model variations b) How and from what time points clones were selected for sequencing - do they represent random or best growing clones c) what assumptions and calculations underlies extraction of generations, CCD and growth rates and the equation of growth rate with fitness; these are key parameters and the supplementary space is large. d) variant calling is not described, neither in terms of parameter settings? Filters? How many reads needs to support a position to be called a mutation? The python script for cnv calling is thinly described and not obviously made available. Table S2 that describes the major parameters in the experiment requires substantial guessing to understand what the different columns actually mean. The S1 materials and methods could be expanded substantially here, as most other sections. As an example, Monomethyl Succinate 1 shows a decrease of growth rate over the course of the experiment, yet it is claimed that it took 188 generations of optimization. What does this mean? Did it take 188 generations to reduce the growth rate, or is this simply the length of the experiment? In the latter case, I don't see the biological relevance of this information. There is no "optimization" so it can hardly be talked about as such. In the case of D-2-Deoxyribose 1, in which observations are

the basis for discussion of the rsbK N20Y mutation, being targeted for transcriptomics, the large deletion of 171 genes as well as being part of validating the model with empirical data, it appears from Table S2 that this is done with a single biological sample? What happened to the replication in this environment? If this is lost, it seems like there are several conclusions based on an n=1 observation.

7. To put this research in the perspective of evolution in the wild, the authors highlight the mutation rsbK N20Y as being found in other bacteria sharing the same environmental factor. The line of evidence here seems to be i) The mutation exists in at least one clone and remains in clones extracted from a population of experimentally evolved E.coli adapted to D-2-deoxyribose. ii) The variant exists in pathogenic bacteria able to grow in D-2-deoxyribose. I miss a crucial piece of evidence - that the mutation has a causal relationship with the phenotype. The evidence presented in this manuscript according to Table S3 is suggesting that it's not, at least not as a single mutation. More generally, the choice of one example to test a generally stated point - is a questionable practice. Any other chosen example would potentially suggest something different. As written, the text implies that the authors are trying to answer a generally phrased question. That should not be done with a sample size of 1.

Reviewer #2:

This paper combines modeling and experiments to investigate if and how enzyme promiscuity opens routes for evolutionary innovation. Specifically, E. coli populations are gradually evolved to grow on carbon sources that do not allow growth of wild-type cells, after which the adaptive mutations are mapped and confirmed.

Overall, this study offers an interesting and valuable overview of different evolutionary routes that allow adaptation to novel carbon sources, including changes in coding regions of metabolic and regulatory proteins, as well as mutations that change expression levels of key genes. I generally like the experimental setup and I think that it is a solid study. That said, I do think that the authors should better acknowledge the (inherent and unavoidable) limitations of their study instead of over-generalizing the conclusions.

I have two major points that I would like to raise.

Firstly, I do not understand (or agree with) the question the paper claims to answer. In the introduction, the authors write "it remains poorly explored how and at what evolutionary stages side activities contribute to adaptation.... The initial "innovation" step" or the subsequent "optimization" step. I do not agree with that statement, because, as far as I understand, the contribution of side-activities (or "promiscuity") is conceptually quite simple. Instead of having to develop a completely novel, previously non-existing function or activity, organisms can simply optimize an existing side-activity. So, the existence of this side activity takes the actual "innovation" step away (at least for that specific function, it is of course possible that other functions or regulatory steps still require innovation, but those are molecularly independent). In fact, the realization that such promiscuous activities are everywhere has meant a breakthrough in our understanding of molecular evolution because innovation (without promiscuity) would typically require a very improbably combination of different mutations, with each of the variants along the mutational path maintaining some function that selection can act upon.

Specifically, I do not understand why the authors claim that "Typically, promiscuity has been linked to the innovation phase" and that they demonstrate that "promiscuity can be linked to fitness benefits in both the innovation and optimization phase". Instead, promiscuity bypasses the real innovation phase and allows organisms to immediately start the optimization (of the specific promiscuous function and/or its regulation).

The existence of promiscuous activities means that some basal capabilities of coping in a new environment are already present, even before exposure to that environment. In many cases, this implies that the organisms can already survive in a new environment but require further adaptation to the promiscuous function in order to increase their fitness in the new environment. Of course, In some cases, gene regulation also needs to be optimized to express the promiscuous activity in the novel environment (but that is perhaps not really innovation, and this step can also be considered

orthogonal to the evolution of the function itself).

My best guess is that this confusion arises from the fact that the authors have developed a particular two-phase regime for their experimental evolution experiments, where they call a first phase with mixed carbon sources the "innovation" stage and a second phase the "optimization" phase. While I understand the reasoning behind this argument (during the mixed carbon source period, the cells cannot survive yet on only the "novel" carbon source, so that "innovation" still needs to happen). However, this is a bit arbitrary, as in many cases, even weak promiscuous activity allows survival or even (very) slow growth on the novel carbon source. So where exactly is the end of the innovation period? Just the mere fact that you do not see growth after some (arbitrary) time in a (arbitrary) experimental setup is perhaps not a strong foundation to call this a specific evolutionary phase, no? I would argue that as soon as some molecule already has a trace of an activity, the true innovation has happened.

In other words, I feel that the division between the innovation and optimization stages is not justified and not helpful to understand the adaptive process, and that this is not a good foundation to build the main message of this paper. The good news is that even without trying to answer whether promiscuous activity enables innovation and/or optimization, the study does contribute to our understanding of different routes in evolutionary adaptation, mostly because instead of studying one example, the authors use 5 different carbon sources and repeat each adaptation twice. This results in a nice and unprecedented view on the different possible routes that evolution takes when starting from a promiscuous activity. In some cases, mutations are needed to increase the expression level of the respective promiscuous gene, whereas in others, the promiscuous function can be augmented by structural changes; or a combination thereof. I think it would be useful to plot the exact routes for each of the 5 examples and try to find general themes (perhaps also with some of the previously published examples) that will help to understand and classify other examples in future studies. For example, I am intrigued by the tradeoffs that come with innovation (an enzyme gains novel function B but may lose its primary function A), and this may merit way more attention. The authors find support for "weak tradeoffs", but other studies have shown much stronger tradeoffs; so that would be great to discuss.

(I am of course happy to hear the arguments of the authors to maintain their claims).

Second, some sections of the manuscript are not very clearly written. For example, the whole paragraph that starts at line 253 is difficult to follow and merits a clearer structure with more information. Similarly, Figure 1 could also use more explanation, and for example a better indication of which phase is considered "innovation" (or perhaps more accurately mixed carbon source) and which ones are "optimization" (single novel carbon source). Also, it is not weird that some mutations seem to happen at the beginning (and not the end) of a plateau phase?

Minor:

Line 171: "... in the laboratory MAY indeed BE relevant to evolution in the wild". ("is" is perhaps too strong, and also insert "be")

Line 308: I don't think that it is fair to claim that "this study shows that enzyme promiscuity is prevalent in metabolism and plays a major role in ... innovation". Such claim would require an enormous amount of work to study the role of promiscuity innovation in many different metabolic steps or routes. Again, I feel that the real merit of the study is in confirming that promiscuity can indeed facilitate evolution, and showing a few of the possible routes and tradeoffs.

Reviewer #3:

This study investigates how novel functions evolve in metabolic enzymes. Authors find that computationally reconstructed metabolic networks can predict the evolution of novel function if the promiscuous functions (underground reactions) are plugged in. Most importantly, they use laboratory evolution experiment as opposed to the artificial overexpression employed before (Notebaard et al. 2014). They provide solid evidence for their findings using multiple approaches. Their findings make a strong case that extensive information about the promiscuous activities of proteins, when combined with computational methods, can give a good predictive power.

The findings of this study will be of interest to the evolutionary biologists interested in studying the origin of innovations and I think this paper is suitable for publication in Molecular Systems Biology. Please find my detailed comments below.

- > Separation of results and discussion sections will improve the readability substantially. It will also help the reader if the title of each result section states the actual result clearly.
- > Many sentences, throughout the manuscript, are too long. Widespread use of commas, parentheses, and dashes, to connect different parts of a sentence, is a bit jarring. It would be helpful if the sentences are shorter and in the active voice, wherever possible. At several instances, the tense changes from one sentence to the next, without any apparent reason.
- > Authors have stressed the point that promiscuous enzyme activity is widespread (Lines 308-311). I think the jury is still out on how prevalent this phenomenon is. The examples provided in this study were already outlined in the Notebaart et al. 2014. The USP of this paper, in my understanding, is that the network approach could predict the outcome of experimental evolution. This point should be prominent in the conclusions.

Lines 49-60

- > This section is a bit confusing. I request the authors to provide clear details of what information was used from Notebaart et al. 2014 study and what was done in this study. Additionally, please provide actual numbers rather than a phrase like 'set of underground reactions', throughout the manuscript (other instances include Line 88 - 'sequenced shortly after', Line 207 - 'several mutations', SOM >Laboratory Evolution Experiments > penultimate line > 'growth rate was monitored periodically')

How did authors choose the eight substrates used for the experimental evolution? Were these the only candidates predicted by the network? If not, how many were there and what was the criteria used for choosing these eight (even if it was an arbitrary choice, please mention so)?

Line 50

- > Comma after 'sources'

Line 65

- > Suggested replacement 'cells acquired the ability to grow on the non-native.....' (To match with the following part of the sentence)

Line 68-70

- > The sentence means that the same E. coli culture was adapted to five different non-native carbon sources. Please modify.

Line 90-92

- > 'Strong signs...' Please point to the appropriate table/fig/dataset for this claim.

Line 102

- > If you mean 'Innovation phase', please remove the word 'initial' (it can create a confusion in reader's mind that you are talking about the beginning of the innovation phase in particular). If you do not mean that, more explanation is needed.

Line 109

- > Since authors revisit this issue later, it would help to point out in the bracket that 'see section..' for further details.

Lines 155-172

- > I suggest either substantiating this part or deleting it for two reasons.
 - a. This is not an organic part of 'Underground metabolism accurately predicted the genes mutated during innovation'. And does not seem to be coherent with other two result sections as well.
 - b. Authors assert the importance too strongly (Line 171) with only one example from the natural isolate.

Lines 156-157

- > Commas missing after N20Y and D-2-deoxyribose (assuming that you meant 'The N20Y,....., served as a case study').

Line 177

- > Suggested removal of 'Specifically'.

Line 208

- > 'directly linked to influence enzyme promiscuity'. Remove 'the' and 'of'.

Line 209

- > Please avoid repetition 'optimizing mutations involved with optimization'.

Line 213

- > Please rephrase. 'during after' is probably a typo and 'initial innovation' could confuse the reader.

Line 270-271

> Please give a proper reference or just write 'previous' and cite. '1977 study' makes little sense.

Lines 299-300

> 'Turning to' sounds odd. Please rephrase.

1st Revision - authors' response

8th November 2018

Response to Reviewers

Manuscript #: MSB-18-8462

Reviewer's Comments: Black Author's Comments: Blue Text

First and foremost, we would like to thank the referees for their time, critical evaluation, and insightful review of the manuscript. Each remark given by each reviewer was thoroughly examined. Our response to each remark is presented below. Major changes made to the manuscript are highlighted with blue text. We chose to address Reviewer #2's comments first during the revisions process, thus that is why that reviewer's comments appears first.

Reviewer #2:

This paper combines modeling and experiments to investigate if and how enzyme promiscuity opens routes for evolutionary innovation. Specifically, *E. coli* populations are gradually evolved to grow on carbon sources that do not allow growth of wild-type cells, after which the adaptive mutations are mapped and confirmed.

Overall, this study offers an interesting and valuable overview of different evolutionary routes that allow adaptation to novel carbon sources, including changes in coding regions of metabolic and regulatory proteins, as well as mutations that change expression levels of key genes. I generally like the experimental setup and I think that it is a solid study. That said, I do think that the authors should better acknowledge the (inherent and unavoidable) limitations of their study instead of over-generalizing the conclusions.

We appreciate the comments provided by this reviewer and have made changes to the manuscript in the hopes of clarifying the issues that were brought up. In regards to better acknowledging the limitations of our study and over-generalizations, we have edited the language in the conclusions section and throughout the paper to have fewer generalizing statements. We have also added a paragraph to the conclusions section that specifically outlines some of the drawbacks/limitations of our study and proposes future work that would enhance the highlighted findings.

I have two major points that I would like to raise.

Firstly, I do not understand (or agree with) the question the paper claims to answer. In the introduction, the authors write "it remains poorly explored how and at what evolutionary stages

side activities contribute to adaptation.... The initial "innovation" step" or the subsequent "optimization" step. I do not agree with that statement, because, as far as I understand, the contribution of side-activities (or "promiscuity") is conceptually quite simple. Instead of having to develop a completely novel, previously non-existing function or activity, organisms can simply optimize an existing side-activity. So, the existence of this side activity takes the actual "innovation" step away (at least for that specific function, it is of course possible that other functions or regulatory steps still require innovation, but those are molecularly independent). In fact, the realization that such promiscuous activities are everywhere has meant a breakthrough in our understanding of molecular evolution because innovation (without promiscuity) would typically require a very improbably combination of different mutations, with each of the variants along the mutational path maintaining some function that selection can act upon.

Specifically, I do not understand why the authors claim that "Typically, promiscuity has been linked to the innovation phase" and that they demonstrate that "promiscuity can be linked to fitness benefits in both the innovation and optimization phase". Instead, promiscuity bypasses the real innovation phase and allows organisms to immediately start the optimization (of the specific promiscuous function and/or its regulation).

The existence of promiscuous activities means that some basal capabilities of coping in a new environment are already present, even before exposure to that environment. In many cases, this implies that the organisms can already survive in a new environment but require further adaptation to the promiscuous function in order to increase their fitness in the new environment. Of course, in some cases, gene regulation also needs to be optimized to express the promiscuous activity in the novel environment (but that is perhaps not really innovation, and this step can also be considered orthogonal to the evolution of the function itself).

My best guess is that this confusion arises from the fact that the authors have developed a particular two-phase regime for their experimental evolution experiments, where they call a first phase with mixed carbon sources the "innovation" stage and a second phase the "optimization" phase. While I understand the reasoning behind this argument (during the mixed carbon source period, the cells cannot survive yet on only the "novel" carbon source, so that "innovation" still needs to happen). However, this is a bit arbitrary, as in many cases, even weak promiscuous activity allows survival or even (very) slow growth on the novel carbon source. So where exactly is the end of the innovation period? Just the mere fact that you do not see growth after some (arbitrary) time in a (arbitrary) experimental setup is perhaps not a strong foundation to call this a specific evolutionary phase, no? I would argue that as soon as some molecule already has a trace of an activity, the true innovation has happened.

We agree it was likely too strong to state that it remains 'poorly explored' how enzyme side activities contribute to adaptation, and we have adjusted this language in the introduction. We believe that (as this reviewer has commented) some confusion about the definition of innovation and optimization did arise with the way the manuscript used these terms. It seems that there were three ways to define/interpret innovation: 1) innovation stage in regards to the

experimental setup, 2) innovation in regards to the existence of an enzymatic side activity, and 3) innovation in regards to achieving the observable phenotype of growth on the new carbon source and those mutations that enabled that phenotype. We addressed this confusion by changing the names of the experimental stages, as well as clearly defining innovation in the introduction specifically related to 3) and clarifying our intentions of the language used throughout the text.

We understand, as this reviewer states, that the fundamental enzymatic innovation already exists by the mere presence of a secondary, promiscuous activity; however, our focus was on the innovation mutation events that aided in selecting for these activities and that were causal in resulting in the observable adaptive phenotype of growth on the new carbon source. The term 'innovation' in this regards has previously been used and we cite this term along with the 'optimization' term from the 2013 Barrick and Lenski Nature Reviews Genetics article, as we did not want to create even more terminology. In this sense, the role of enzyme promiscuity has indeed largely been linked to providing an advantage in adapting to new ecological niches/ non-native environments. We first decided to explore the mutation events that were required for that initial ability to grow on the non-native carbon source (growth being defined as detectable in our experimental set-up which was limited by the optical density plate reader and persistent as the cells were passed to new flasks). Looking at mutation causality for an observable and reproducibility measurable phenotype is not then arbitrary in this sense and is a common practice for examining evolution experiments. We followed up this analysis by then also examining those mutations that helped to enhance growth in the new environment which is marked by a period of incremental benefits in growth fitness.

In other words, I feel that the division between the innovation and optimization stages is not justified and not helpful to understand the adaptive process, and that this is not a good foundation to build the main message of this paper. The good news is that even without trying to answer whether promiscuous activity enables innovation and/or optimization, the study does contribute to our understanding of different routes in evolutionary adaptation, mostly because instead of studying one

example, the authors use 5 different carbon sources and repeat each adaptation twice. This results in a nice and unprecedented view on the different possible routes that evolution takes when starting from a promiscuous activity. In some cases, mutations are needed to increase the expression level of the respective promiscuous gene, whereas in others, the promiscuous function can be augmented by structural changes; or a combination thereof. I think it would be useful to plot the exact routes for each of the 5 examples and try to find general themes (perhaps also with some of the previously published examples) that will help to understand and classify other examples in future studies. For example, I am intrigued by the tradeoffs that come with innovation (an enzyme gains novel function B but may lose its primary function A), and this may merit way more attention. The authors find support for "weak tradeoffs", but other studies have shown much stronger tradeoffs; so that would be great to discuss.

(I am of course happy to hear the arguments of the authors to maintain their claims).

We have made an effort to make our analysis more logical in focusing on the adaptive trends in each of the cases, rather than focused around two experimental stages as this reviewer brings up. Please make note of the sub-heading name changes and some text re-arrangement and text additions that have been made to the results and discussion section. For example, when looking at the growth-optimizing mutations after determining mutation causality, we highlighted three distinct mutation trends that were observed in our experimental results (structural mutations linked to shifting substrate affinities, regulatory element mutations linked to increased expression of underground activities, and large duplication/deletion events). This restructuring hopefully achieves what this reviewer has suggested in better plotting the various evolutionary routes observed in these distinct experiments. As far as an in depth discussion of the tradeoffs that come with innovation, we feel this avenue of research would require many more follow-up experiments and enzyme kinetics assays that would significantly expand the length of the manuscript and delay the reporting of these findings; however, we did make changes to the conclusion to suggest future work to be done utilizing some of these case studies as a starting point for that type of analysis.

Second, some sections of the manuscript are not very clearly written. For example, the whole paragraph that starts at line 253 is difficult to follow and merits a clearer structure with more information. Similarly, Figure 1 could also use more explanation, and for example a better indication of which phase is considered "innovation" (or perhaps more accurately mixed carbon source) and which ones are "optimization" (single novel carbon source).

We have made changes to the names of the experimental phases in Fig 1 to have them describe the experiment setup utilized explicitly. Further, we have reviewed the paragraph in question and edited it, breaking it up into multiple paragraphs and clarifying the denser language used to hopefully make the message clearer. Please see the changes made.

Also, is it not weird that some mutations seem to happen at the beginning (and not the end) of a plateau phase?

We did not sequence continuously along every data point and saved stocks periodically, thus we were limited to certain flasks which could be sequenced and analyzed. We have added a sentence on page 5 that clarifies why fixing mutations are assumed to be present throughout the plateau phase along with a reference which provides justification for this assumption using our experimental setup.

Minor:

Line 171: "... in the laboratory MAY indeed BE relevant to evolution in the wild". ("is" is perhaps too strong, and also insert "be")

Thank you for this comment. Since this example appeared to fall somewhat out of the scope of the main message of this paper after the reorganization and was confusing to the other reviewers as well, we decided to move it to supplementary material and change the wording, as suggested.

Line 308: I don't think that it is fair to claim that "this study shows that enzyme promiscuity is prevalent in metabolism and plays a major role in ... innovation". Such claim would require an enormous amount of work to study the role of promiscuity innovation in many different metabolic

steps or routes. Again, I feel that the real merit of the study is in confirming that promiscuity can indeed facilitate evolution, and showing a few of the possible routes and tradeoffs.

We appreciate this comment that we might have made an overgeneralization with this statement, and have made an effort to re-word and expand the conclusions section to better clarify and summarize our findings. Please make note of the added/changed text in the conclusions section.

Reviewer #1:

Guzman et al combines experimentation and computation to understand: a) how the capacity to utilize novel energy/carbon sources using promiscuous enzymes evolves b) how well solutions can be predicted based on flux balance metabolic models. The paper's contribution to the first issues is encapsulated in the statements: "we demonstrate that enzyme promiscuity can be linked to fitness benefits in both the innovation and optimization stages of adaptive evolution." The contribution to the second issue is summarized by: "genes underlying the phenotypic innovations were accurately predicted by genome-scale model simulations of metabolism with enzyme promiscuity."

The authors nicely integrate experimental and computational techniques, present an extensive body of work, and experiments and analysis, with a few exceptions, were well performed. If revised, the paper is likely to be well received by sections of the experimental evolution and metabolic modelling communities; I am not entirely convinced about the size of these sections nor that it will attract a broader readership.

Specific concerns:

1. The authors emphasize the conclusion: "we demonstrate that enzyme promiscuity can be linked to fitness benefits in both the innovation and optimization stages of adaptive evolution". They present this as a major finding. I am not sure that this finding represents a substantial advancement of evolutionary biology. Why would enzyme promiscuity be beneficial in only one of these stages? Is that a theory-driven proposition? What is the rationale? Indeed, the authors do not further discuss the significance of this finding in the conclusions, suggesting that they are not convinced that this is a significant finding either.

To address this comment, which is in line with Reviewer #2's major critique, we have restructured the manuscript away from the innovation and optimization segmented format to focus on the major evolutionary themes from looking at the multiple cases presented in the manuscript. We have also expanded the conclusions to discuss the major findings highlighted in this new manuscript structure.

Second, it is not clear to me why optimizing mutations would not emerge in what in the design is called the innovation phase.

Thank you for this comment. We realize that our terminology of innovative and optimizing could have been misleading. Please see our response to Reviewer 2. We have made an effort to better define innovation and optimization as well as renamed the evolution experimental stages to separate these terms. In our manuscript we refer to optimizing mutations as those that improve the growth rate/fitness once growth on the non-native carbon source is established. We agree that it is possible that multiple mutations could have fixed during the weaning phase beyond those that enabled growth on the non-native substrate (beyond those that were innovative mutations as verified in the knock-in experiments). These mutations could then also be termed optimizing mutations or possibly neutral mutations, beyond those that are causal. In our experiments, however, we did not observe many mutations that fit this scenario. One exception was the case of mutations observed for the D-2-deoxyribose experiment. In this case we were unable to pin down a single causal mutation or innovative mutation, but rather it was possible that the two or three mutation events observed (rbsK mutation, rbsR mutation, or large duplication events) could have been required for growth. We have added some statements to the text to hopefully clarify this exception.

Third, given that the allele frequencies are not tracked over time and clone dynamics are unknown, it seems dangerous to make assumptions about when mutations first emerge (pre-experiment,

innovation and optimization phase), in what order, and to what degree their fitness benefits manifest in these phases.

Indeed, allele frequencies were not monitored continuously throughout the experiment, thus we have restricted our comments about the dynamics of the evolution to only those justifiable by analyzing the clones which were harvested along the course of the evolution. Clearly, there was an order to mutation occurrence given that specific sets of mutations repeatedly appeared earlier during the course of the evolutions. For example, *fucR* in the D-arabinose evolution, *dmlA* in the m-tartrate evolution and a number of additional cases listed in Table 1.

Fourth, is there really a clear qualitative functional distinction between "innovative" and "optimizing" mutations here? Can the authors reject that sequencing of mutations is not only a matter of chance, local mutation rates, or small differences in selection coefficients? The authors classify the mutation *rsbK* N20Y as an innovative mutation in Table 1, yet when reconstructed it does not seem to be sufficient for growth even after a week, according to Table S3. It strikes me as surprising since the innovative mutations were discovered initially by the innovation test lasting for only 3 days. The authors comment on this finding arguing that it's a complex scenario requiring additional regulatory mutations, but isn't that very close to the definition of an optimizing mutation? Why is not the reconstruction of this regulatory mutation done and tested? According to Figure S1, it appears that the first sampling for sequencing where the *rsbK* mutations are found is performed after substantial adaptation. Without

phenotyping both *rsbK* mutations how do the authors reject the possibility that *rsbK* N20Y is an "optimizing" mutation in this case? I think this is a striking example of the seemingly arbitrary boundary between "innovative" and "optimizing" mutations from a biological perspective. In summary - I am not yet convinced by the clear separation the authors want to make between "innovation" and "optimization" and find it artificial and in danger of being near entirely subjective.

We appreciate this comment and believe that the changes we have made to the manuscript will clarify the confusion surrounding the 'innovative' and 'optimizing' terms used. First off, there appears to be a confusion with the title of Table 1. We meant that those mutations listed were linked to the innovative growth phenotype without meaning that they were had been deemed causal on their own. In response to this reviewer's comments about the *rsbR* mutation, as seen in Table S3, we did indeed examine the regulatory mutation in *rsbR* (the entry directly above the *rsbK* mutation) and found that the regulatory mutation on its own was also not sufficient for growth. Indeed, we could not reject the possibility that any of the mutations observed for the D-2-deoxyribose case were optimizing or even neutral. We apologize if that was the interpretation that this reviewer perceived from reading these results. We have changed the title of Table 1 in hopes of clarifying this as well as edited the text surrounding the D-2-deoxyribose case.

2. In the conclusions, the authors' introduce two alternative key conclusions: "First, side activities contributed to the establishment of novel metabolic routes that enabled or improved the utilization of a new nutrient source. Second, suppression of an undesirable underground activity that diverted flux from a newly established pathway conferred a fitness benefit." The first statement is well supported by the data but not entirely novel in itself. More problematically, I don't feel that they have sufficiently well supported the second statement.

We appreciate this comment and have expanded and restructured the conclusions section to better summarize our findings. The second statement is no longer included in the conclusion and we have followed this reviewer's advice in toning down the stronger claims made by our analysis of the large deletion. Please see our response to the next comment below.

The authors postulate that a large deletion of 171 genes present at unknown frequencies in endpoint populations is the driver of adaptation at previous time points. It is unclear what the empirical support driving this conclusion is as there is no recurrence, no reconstructions and no time course sequencing. How do the author's reject the null hypothesis of the mutation being neutral? Further, the authors conclude that *AldA* deletion is likely the driver, because in the founder *AldA* had the most highly expressed metabolic transcript under these conditions. There are multiple problems with this inference: the tenuous link between transcripts and protein activity, between high protein

activity and deleterious effects of the protein, the assumption that the driver mutation must be metabolic rather than regulatory, the assumption that founder cell state reflects the cell state after innovation, and the assumption that none of the 171 deleted genes affect fitness individually or in combination (it seems rather unlikely that these would all

be neutral). Several things could and should have been done, with deleting AldA from an innovated, not optimized clone and inserting AldA into an optimized clone being the most important.

We appreciate these comments and have followed this up by conducting some experiments to further investigate and test the hypothesis set by our AldA analysis. We did conduct gene deletion experiments, deleting the *aldA* gene from two clones isolated prior to the large deletion event (those highlighted on the D-2-deoxyribose trajectory on Appendix Figure S1A and whose mutations are described in Dataset S1). Growth was monitored for these *aldA* deletion mutants compared to those not containing the deletion and we did not see an improvement in the growth rate, which we would have expected based on our predictions. Thus, we have changed the text of this section in the manuscript, made changes to Figure 4, and moved it to the supplement. Furthermore, we have also pointed out that other factors may be influencing the observed growth fitness improvement, such as a smaller deletion event found in *rbsB* and in the intergenic region upstream of *rbsK*.

Finally, the authors support their statement with an FBA, stating that conversion of acetaldehyde to acetate has negative effects on growth rate. How large was the effect?

The effect was quantified in figure S10B with the stated model parameters in the methods. This setup was the basis for a specific hypothesis of how the removal of a gene could alter an observed growth-rate/fitness. The size of the effect as predicted by the model was displayed in Figure S10B, however, experimental removal of the *aldA* gene did not result in an improved growth rate in clones isolated prior to the deletion. As you can read now in the edited version of the text, we explain that it is possible that other mechanisms are at work here and rather than place so much focus on *aldA* (one gene out of the 171 genes deleted) we have broadened our analysis to examine the other metabolic genes in the large deletion.

What is the precision and accuracy of the model?

The precision and accuracy of constraint based models is discussed and analyzed in depth at the point of release of each model. For this work we utilized the iJO1366 model (as stated in the methods and in the manuscript). We would invite the reviewer to review the publication associated with this model (Orth et al. 2011) along with a follow-up paper regarding gap-filling techniques (Orth and Palsson 2012) to better familiarize themselves with these models and procedures. We do not believe that it is common practice to discuss the precision and accuracy of well-established models when this information is already readily available and has been cited in our manuscript.

What other fluxes have negative effects on growth rates in the model? Larger negative effects on growth rates?

The field of constraint based modeling is well established and systematic analyses on gene knockouts and knock down simulations have been extensively characterized. Details on negative loss of function deletions and mutations, which can be viewed as shadow prices, has been previously reported and can be found here: (Palsson 2015). It does not seem warranted to detract from the message of the manuscript with repeating such an analysis here.

Are any of these genes in the deleted segment? What is the chance probability that any one of the genes associated with negative effects on the growth rates in the model would occur in a 171 gene deletion? Are there additional enzymes catalyzing the reaction in *E. coli*? *aldA* is claimed to be most significantly differentially expressed among the metabolic genes, but are there any other metabolic genes among the differentially expressed genes? How does the modeling look for those genes compared to *aldA*? I cannot help feel that the evidence presented in 2.4. is quite circumstantial - the conclusion is not well supported by data, as stands.

We appreciate the reviewer's concern for this analysis and have taken several additional steps to more comprehensively analyze the large deletion observed in the D-2-deoxyribose evolution experiment. We have removed our initial emphasis on the *aldA* gene and expanded our analysis to all metabolic genes (genes included in our model of metabolism) located within the deletion region.

We have found that there were other metabolic genes (18 genes) that were differentially expressed along with *aldA* and flux variability analysis conducted on the reactions associated with the 44 metabolic genes located in 171 gene deletion region showed that none of these are necessary to achieve an optimal growth solution. We have thus moved Figure 4 to the supplement (Fig. S10) and added a dataset that provided more details about the 44 metabolic genes located within this deletion region (Dataset EV3). Furthermore, the text originally emphasizing the role of *aldA* in growth optimization has been completely re-written to reflect the new analysis. We believe this analysis will satisfy the issues brought up by the reviewer.

3. I am surprised and a little concerned about the data presented in Table S3. The authors very nicely reconstruct many of the candidate innovative mutations in a founder background - excellent and laborious job. But they do not do a proper quantification of the effect of the mutation on growth rates in the growth conditions of interest. Instead they do a qualitative, subjective and presumably quite error prone assay "time to visual first growth". This rounded to "full days". This is not entirely convincing. I would like to see the growth rates and I would like to see the growth rates converted into selection coefficients (the authors equate growth rates with fitness so this should be a straightforward transformation). It's unclear how a week was selected as the duration for the experiments underlying Table S3. As understood from SI material and methods the classification and discovery of an "innovative" mutation is visible growth after three days. It is quite remarkable that none of the mutants tested for Table S3 grows to visible growth in <3 days. Does this mean that none of the mutations tested for Table S3 actually were "innovative" mutations? Ideally, I would like to see some kind of perspective on whether these mutations realistically could explain the type of adaptation kinetics observed. In this context, I am a little concerned by statements such as (in the abstract). "After as few as 20 generations, the evolving populations repeatedly acquired the capacity to grow on predicted non-native substrates". First, the statement is ambiguous - it is unclear from where it is taken, how the number is calculated and what the authors really mean, explicitly? At a casual reading, 20 generations seems to imply very fast adaptation. Can the reconstructed mutations, which seem very weak, really explain that? It seems highly unlikely to me, especially assuming that the

population starts out without standing variation. According to Table S2, 20 generations is nowhere to be seen. Something is amiss here.

We apologize for the confusion. The goal of the individual mutant growth test was to look for growth or no growth, within a reasonable time frame - which we set as a week, without any prior adaptation on the non-native substrate. Conducting a more extensive growth characterization detailing growth rates to explain adaptation kinetics of each clone/experiment could be informative, but the goal was to make a binary growth or no growth call, and then to confirm that the strain that grew in each experiment was the inoculated strain using sequencing of the intentionally mutated region. We agree that the timeframe of a week is an arbitrary choice, however, growth/no growth tests (as have been reported in many studies, e.g. Keio collection growth studies and the like) inherently set 'arbitrary' timeframes for making growth calls as they are arguably unavoidable. Thus, table S3 was modified to simply state if growth was observed or not.

Given the number of clones and the noted already laborious effort to construct them, we agree that these clones could form the basis for such a detailed comparative study of adaptation kinetics and is something we are considering.

In response to the expectation of seeing growth in less than 3 days, this is likely due to the fact that these individual mutants were washed twice prior to inoculation and thus the cells underwent a likely a temperature shock along with the perturbation from centrifugation. Further, with the cells being washed twice, a small inoculation population was used. Therefore, it was not surprising to us that the mutants required more time to observe growth. The washing step was detailed in the methods section. We would also like to note that each of the cultures that showed growth were sequenced to confirm they were indeed the expected strain and no additional mutations were within the examined region.

As far as the ambiguity mentioned regarding the 20 generations mentioned, we apologize if this was not more apparent in Table S2. We should have added the word 'approximately' in front of this

number, which we have now added. We have made adjustments to the Table S2 headings in hopes of clarifying this confusion. The 20 generations comes from a rounding of the monomethyl succinate replicate 1 experiment (21 generations) and the m-tartrate replicate 1 experiment (16 generations).

4. The paper feels like a somewhat hastily composed and slightly disorganized product. Too many, somewhat disconnected follow-up threads joined together with an unclear rationale and sequencing and division into paragraphs of very uneven length. Typos, sentences with unclear syntax and grammar and duplicated refs that should have been weeded out before submission. Figure legends that in many cases lack key information on error bars, type and number of replicates and tests behind reported p-values. Mutations that often are annotated only as the affected gene. References to the wrong display item, or statements not supported by references

to display items. It feels like a not sufficiently well crafted final product and the authors should perhaps go back to the drawing board.

To address this critique, we have gone through additional rounds of editing and carefully revised the text to review and correct the grammar and any reference errors. We apologize for any confusion this may have caused when reading the manuscript.

5. The conclusion "genes underlying the phenotypic innovations were accurately predicted by genome-scale model simulations of metabolism with enzyme promiscuity" is probably well supported. However, the formal statistics underlying this conclusion is unclear (F1 score?) and the authors do not explain the erroneous prediction for D-arabinose. While the model missed the fuc operon associated genes, it is not clear why it did the prediction it did and why the predicted gene was not empirically validated. In Fig S2, which is supporting the main statement about how well the authors modeling findings can be experimentally observed, the genes *fucI*, *fucK*, *fucA* are claimed to be experimentally observed. I would expect to find these genes in either Fig S1 or in Table 1. I do not. Since these genes make up 100% of one of the cells in the contingency table it's surprising that I cannot find references to any of these genes in the rest of the paper in terms of results.

We apologize if this information was unclear, however it was included in the manuscript. Specifically, we would direct you to the last sentence in the first paragraph under section "Modeling with underground metabolism accurately predicted key genes mutated during laboratory evolution experiments". This sentence does indeed list the proteins *FucI*, *FucK*, and *FucA*, which were related to the observed *fucR* mutation which is listed in Fig. S1 and Table 1. Furthermore, these genes are also listed in the pathway map in Fig. 3C and the metabolism of D-arabinose is discussed in more depth later on in the paper under section "Mutations in regulatory elements linked to increased expression of underground activities: D-arabinose evolution". We added a statement where these genes are first mentioned in the text to reference the section that more thoroughly explores D-arabinose metabolism in order to hopefully make it clearer where these specific pieces of information can be located.

6. The M&M section and supporting figs is too concise and in some cases do not allow for a good evaluation of results. This includes a) explanations for choices, assumptions, and parameters estimates in the model and the robustness of conclusions to model variations b) How and from what time points clones were selected for sequencing - do they represent random or best growing clones c) what assumptions and calculations underlies extraction of generations, CCD and growth rates and the equation of growth rate with fitness; these are key parameters and the supplementary space is large. d) variant calling is not described, neither in terms of parameter settings? Filters? How many reads needs to support a position to be called a mutation? The python script for cnv calling is thinly described and not obviously made available. Table S2 that describes the major parameters in the experiment requires substantial guessing to understand what the different columns actually mean. The S1 materials and methods could be expanded substantially here, as most other sections. As an example,

Monomethyl Succinate 1 shows a decrease of growth rate over the course of the experiment, yet it is claimed that it took 188 generations of optimization. What does this mean? Did it take 188 generations to reduce the growth rate, or is this simply the length of the experiment? In the latter

case, I don't see the biological relevance of this information. There is no "optimization" so it can hardly be talked about as such. In the case of D-2-Deoxyribose 1, in which observations are the basis for discussion of the rsbK N20Y mutation, being targeted for transcriptomics, the large deletion of 171 genes as well as being part of validating the model with empirical data, it appears from Table S2 that this is done with a single biological sample? What happened to the replication in this environment? If this is lost, it seems like there are several conclusions based on an n=1 observation.

We apologize if this reviewer felt that the methods section were not sufficient to reproduce the results presented in the manuscript. We have revisited several sections in the methods to add details for clarification (please see the highlighted text and changes made in Materials and Methods). In regards to the genome-scale model assumptions, parameters, and robustness assessments, we would like to clarify to this reviewer that this study was largely based on prior modeling results published and fully described in the Notebaart et al. 2014 PNAS publication. Furthermore, the methods that were used in this publication to analyze the effects of gene deletions and pFBA/FVA analyses are now more fully described in the methods and the necessary publications are cited. For more references that discuss the types of assumptions and limitations of these predictions the reviewer may want to look at these publications: (Orth and Palsson 2012; Guzmán et al. 2015; Ibarra et al. 2003; Orth et al. 2011). Since this is predominantly an experimental paper, we do not feel it is appropriate or necessary to reiterate the analysis of previous work.

To address the comments regarding evolution parameters we have expanded upon how calculations were made in the Materials and Methods. In regards to Table S2, we have adjusted the names of the columns to help clarify what was meant by the parameters included in this table. Indeed, the number of generations shown under the column "Generations in Static Phase" represents the number of generations that the evolving populations underwent serial passage in exponential growth, evolving to select for the cells with the highest growth rates. Although, as mentioned by the reviewer, the monomethyl succinate 1 evolution experiment did not result in an improvement of the growth rate, it was subjected to 188 generations of this type of selection and we felt it necessary to include this information for consistency. As mentioned before, we no longer refer to this phase as the 'optimization' phase, so hopefully this will satisfy this reviewers concerns for the inclusion of such information.

Finally, as described in the main text of the manuscript, all evolution experiments were conducted in duplicate for each individual growth substrate. For the D-2-deoxyribose experiments, however, only one of the two replicate experiments was successfully weaned onto the non-native substrate and subsequently growth-optimized during the static environment phase of the evolution. We agree that if we were making claims about the statistical significance of the set of possible mutations required for growth on D-2-deoxyribose, then an n=1 biological

replicate would not be sufficient. In our work, however, we are stating that the evolution of E. coli K-12 MG1655 to grow on D-2-deoxyribose is possible (which is sufficient to say with a single replicate) and this result is novel, to our knowledge. We then take additional steps to examine the mutational events that occurred in that single evolving population. Future experiments with many more replicates could then assess claims regarding what the wider range of mutational possibilities is for evolution on this substrate. This, however, was not the focus of our work so we disagree with this reviewer's assessment.

7. To put this research in the perspective of evolution in the wild, the authors highlight the mutation rsbK N20Y as being found in other bacteria sharing the same environmental factor. The line of evidence here seems to be i) The mutation exists in at least one clone and remains in clones extracted from a population of experimentally evolved E.coli adapted to D-2-deoxyribose. ii) The variant exists in pathogenic bacteria able to grow in D-2-deoxyribose. I miss a crucial piece of evidence - that the mutation has a causal relationship with the phenotype. The evidence presented in this manuscript according to Table S3 is suggesting that it's not, at least not as a single mutation. More generally, the choice of one example to test a generally stated point - is a questionable practice. Any other chosen example would potentially suggest something different. As written, the text implies that the authors are trying to answer a generally phrased question. That should not be done with a sample size of 1.

We appreciate your comments here. Since this example appeared to fall somewhat out of the scope of the main message of this paper after the reorganization and was confusing to the other reviewers as well, we decided to state the finding as an observation with potential relevance, but state that it is a single observation. We have also moved the paragraph to supplementary material.

Reviewer #3:

This study investigates how novel functions evolve in metabolic enzymes. Authors find that computationally reconstructed metabolic networks can predict the evolution of novel function if the promiscuous functions (underground reactions) are plugged in. Most importantly, they use laboratory evolution experiment as opposed to the artificial overexpression employed before (Notebaard et al. 2014). They provide solid evidence for their findings using multiple approaches. Their findings make a strong case that extensive information about the promiscuous activities of proteins, when combined with computational methods, can give a good predictive power.

The findings of this study will be of interest to the evolutionary biologists interested in studying the origin of innovations and I think this paper is suitable for publication in Molecular Systems Biology. Please find my detailed comments below.

> Separation of results and discussion sections will improve the readability substantially. It will also help the reader if the title of each result section states the actual result clearly.

We appreciate this suggestion and have decided to maintain the current structure of results and discussion; however, we have made significant changes to the section headings and restructured the sections to hopefully make the work clearer. We also expanded our conclusions section to better summarize our findings and point out study limitations and potential future work.

> Many sentences, throughout the manuscript, are too long. Widespread use of commas, parentheses, and dashes, to connect different parts of a sentence, is a bit jarring. It would be helpful if the sentences are shorter and in the active voice, wherever possible. At several instances, the tense changes from one sentence to the next, without any apparent reason.

To address this critique, we have gone through additional rounds of editing and carefully revised the text to review and correct the grammar and any reference errors. We apologize for any confusion this may have caused when reading the manuscript.

> Authors have stressed the point that promiscuous enzyme activity is widespread (Lines 308-311). I think the jury is still out on how prevalent this phenomenon is. The examples provided in this study were already outlined in the Notebaard et al. 2014. The USP of this paper, in my understanding, is that the network approach could predict the outcome of experimental evolution. This point should be prominent in the conclusions.

We appreciate this comment. We have made some adjustments throughout the manuscript to hopefully tone down any overgeneralizations we may have made. However, the comments of Reviewer #2 seem to contradict this belief that enzyme promiscuity is not widespread, and we have made an attempt to balance these conflicting statements by explicitly stating limitations. Finally, we have expanded our conclusions section and specifically pointed out the notoriety of being able to predict the outcome of experimental evolution.

Lines 49-60

> This section is a bit confusing. I request the authors to provide clear details of what information was used from Notebaard et al. 2014 study and what was done in this study. Additionally, please provide actual numbers rather than a phrase like 'set of underground reactions', throughout the manuscript (other instances include Line 88 - 'sequenced shortly after', Line 207 - 'several mutations', SOM >Laboratory Evolution Experiments > penultimate line > 'growth rate was monitored periodically')

Thank you for bringing this to our attention. We have added the details mentioned, editing the main text, methods section, as well as Table S1. Please note the changes made.

How did authors choose the eight substrates used for the experimental evolution? Were these the only candidates predicted by the network? If not, how many were there and what was the criteria used for choosing these eight (even if it was an arbitrary choice, please mention so)?

We have expanded the sentence in the first paragraph of the results and discussion section to more explicitly state our selection criteria for these carbon sources (substrate cost, availability, and solubility properties).

Line 50

> Comma after 'sources' We have made this change.

Line 65

> Suggested replacement 'cells acquired the ability to grow on the non-native.....' (To match with the following part of the sentence)

We have adjusted the wording of this sentence.

Line 68-70

> The sentence means that the same E. coli culture was adapted to five different non-native carbon sources. Please modify.

We have modified this statement to state that each carbon source was analyzed individually.

Line 90-92

> 'Strong signs...' Please point to the appropriate table/fig/dataset for this claim.

Thank you for bringing this up. We have made the change to point out the appropriate tables and figures for this.

Line 102

> If you mean 'Innovation phase', please remove the word 'initial' (it can create a confusion in reader's mind that you are talking about the beginning of the innovation phase in particular). If you do not mean that, more explanation is needed.

Thank you for this comment; we have deleted 'initial'.

Line 109

> Since authors revisit this issue later, it would help to point out in the bracket that 'see section..' for further details.

We have made this adjustment.

Lines 155-172

> I suggest either substantiating this part or deleting it for two reasons.

a. This is not an organic part of 'Underground metabolism accurately predicted the genes mutated during innovation'. And does not seem to be coherent with other two result sections as well. b. Authors assert the importance too strongly (Line 171) with only one example from the natural isolate.

Based on this comment as well as those received from the other reviewers, we have moved this section to supplemental material.

Lines 156-157

> Commas missing after N20Y and D-2-deoxyribose (assuming that you meant 'The N20Y,...., served as a case study').

We have made this change.

Line 177

> Suggested removal of 'Specifically'.

This part of the paper was restructured and that phrase is no longer included.

Line 208

> 'directly linked to influence enzyme promiscuity'. Remove 'the' and 'of'. We have reworded this sentence.

Line 209

> Please avoid repetition 'optimizing mutations involved with optimization'. We have fixed this.

Line 213

> Please rephrase. 'during after' is probably a typo and 'initial innovation' could confuse the reader. This sentence was removed.

Line 270-271

> Please give a proper reference or just write 'previous' and cite. '1977 study' makes little sense. We've adjusted this sentence.

Lines 299-300

> 'Turning to' sounds odd. Please rephrase. We reworded this sentence.

2nd Editorial Decision

11th January 2019

Thank you for sending us your revised manuscript. We have now heard back from the two referees who were asked to evaluate your study. As you will see below, both reviewers think that the study has significantly improved as a result of the preformed revisions. However, reviewer #1 still raises some issues, which need to be addressed in a revision.

The more fundamental issues raised are the following:

- i) The model parameters used in the performed analyses need to be provided. Moreover, it should be shown that the model is robust to parameter choice (point 2 of reviewer #1).
- ii) The experiments related to 2-deoxyribose need to be reproduced in order to provide better support for the derived conclusions (points 6 and 7 of reviewer #1).

All other issues raised would need to be convincingly addressed.

REFEREE REPORTS.

Reviewer #1:

The authors have done a good job of responding to my critique, and improving the manuscript This includes substantial experimental effort - much appreciated. However, I do feel that additional modifications are necessary before I can recommend acceptance. Number below relate to my original points.

1: I appreciate the change of terminology, separating the experimental phases into 'weaning' and 'static'. However, the titles of Table 1 and Table 2 still talk about these mutations being associated with "innovative growth phenotypes" versus "optimizing mutations". I maintain that the authors have not causally linked mutations to innovation and optimization phenotypes respectively - they have associated them with the now better named experimental phases. In the rebuttal letter the authors also seem confused. They state:

A) "We meant that those mutations listed were linked to the innovative growth phenotype"

This contradicts their subsequent statement

B) "Indeed, we could not reject the possibility that any of the mutations observed for the D-2-deoxyribose case were optimizing or even neutral."

The authors should be clear in the text and in their tables. They have shown a temporal association of mutations to a phase, not to innovation or optimization and they have not rejected neutrality.

2: I accept that the authors do not wish to give a complete report of model parameters. However, I would welcome a statement specifying that predictions are robust to parameter settings and assumption over a reasonable (for which true value are rarely known). Now I have to take for granted that the authors have made sure that such robustness exist and have not intentionally or accidentally selected a sweet spot were predictions coincidentally match experimental outcomes. Moreover, the authors state:

"We have thus moved Figure 4 to the supplement (Fig. S10)" this is apparently not a correct figure reference, I suppose you mean Fig S8? Please make sure that the figure references in the manuscript point to the correct figures.

5: I apologize if the critique regarding the contingency table in Fig S2 was imprecise. The main

issue is that *FucI*, *FucK* and *FucA* genes were not observed in this work. They relate to a very old, previous observation (LeBlanc & Mortlock, 1971). As presented, it is now included as a result in the statistical analysis. The authors need to be clear. Does the contingency table capture their own data? Or their own data and a thorough meta-analysis of published literature? If the second, a more extensive description of this meta-analysis and the associated statistical problems should be given. Alternatively, they should exclude *FucI*, *FucK* and *FucA* from their statistical analysis and discuss it separately as a previously published finding in the text.

6 and 7: I'm not convinced by the 2-deoxyribose results. As stands the 2-deoxyribose results are entirely based on a single experimental population and clones with reconstructed presumed driver mutation did not grow on 2-deoxyribose. There is no buffer whatsoever against error - mislabelling, mix-up, contamination. Reproducibility is not a luxury item required for proper statistics, it is essential for science. I firmly believe that 2-deoxyribose results should either be reproduced at least once or should be removed from the paper. Moreover, as the 2-deoxyribose results are shaky and the presumed driver not confirmed, an extensive discussion of this mutation and its relation to selection seems premature.

Reviewer #2:

I very much appreciate the efforts of the authors to address my comments, and the clear rebuttal letter and annotated manuscript.

I think that the manuscript is now much clearer. Specifically, it is good that the innovation and optimisation steps are no longer projected as completely separate phases.

I support publication of this revised version.

2nd Revision - authors' response

8th March 2019

Response to Reviewer

Reviewer's Comments: Black

Author's Comments: Blue Text Changes made in manuscript: Blue Text

Reviewer #1:

The authors have done a good job of responding to my critique, and improving the manuscript This includes substantial experimental effort - much appreciated. However, I do feel that additional modifications are necessary before I can recommend acceptance. Number below relate to my original points.

1: I appreciate the change of terminology, separating the experimental phases into 'weaning' and 'static'. However, the titles of Table 1 and Table 2 still talk about these mutations being associated with "innovative growth phenotypes" versus "optimizing mutations". I maintain that the authors have not causally linked mutations to innovation and optimization phenotypes respectively - they have associated them with the now better named experimental phases. In the rebuttal letter the authors also seem confused. They state:

A) "We meant that those mutations listed were linked to the innovative growth phenotype" This contradicts their subsequent statement

B) "Indeed, we could not reject the possibility that any of the mutations observed for the D-2-deoxyribose case were optimizing or even neutral."

The authors should be clear in the text and in their tables. They have shown a temporal association of mutations to a phase, not to innovation or optimization and they have not rejected neutrality.

We apologize if this is still a source of confusion. To clarify and better satisfy this reviewer's concerns, we have renamed the Tables. Please see this change in the manuscript.

The mutations in Table 1 were shown not to be neutral (last paragraph on page 5) as they were reverse engineered into the ancestral strain and determined to be casual for growth. Please see below for additional experimental work relating to the mutations associated with deoxyribose, as they have now been also linked to causality.

2: I accept that the authors do not wish to give a complete report of model parameters. However, I would welcome a statement specifying that predictions are robust to parameter settings and assumption over a reasonable (for which true value are rarely known). Now I have to take for granted that the authors have made sure that such robustness exist and have not intentionally or accidentally selected a sweet spot were predictions coincidentally match experimental outcomes. Moreover, the authors state:

"We have thus moved Figure 4 to the supplement (Fig. S10)" this is apparently not a correct figure reference, I suppose you mean Fig S8? Please make sure that the figure references in the manuscript point to the correct figures.

In our previous work (Notebaart et al. 2014) from which the non-native carbon sources were selected, we predicted that the addition of underground metabolic reactions to the main metabolic network enables growth (compared to metabolic network without underground activities) when the selected non-native carbon sources are the only available carbon sources in the environment. For the modelling, we used flux balance analysis which predicts growth yield as it results in a ratio between the uptake flux (carbon transport reaction) and output flux (growth/biomass reaction). Generally, the only parameters that are set in such models are the uptake fluxes representing the environment. All other (internal) fluxes, which will be predicted, can take up any flux value. Taken together, the outcome of the modelling, i.e., the ratio between uptake and output, can be accepted to be robust against changing the exact uptake flux. This property has been demonstrated in several publications, including a seminal paper on this topic, which we now have cited and added to the manuscript methods (Edwards and Palsson 2000).

Furthermore, we note that in case of all predicted carbon sources studied here, the metabolic network without underground reactions was completely incapable of providing growth owing to a lack of biochemical routes from carbon source to biomass. As such, these can be considered as qualitative predictions that only depend on the structure of the network. We now emphasize this better in the Methods section.

Finally, we wish to emphasize that in our previous work we showed significant agreement of model predictions and genome-wide overexpression screens and now we show significant agreement between enzymes with mutations found experimentally and modelling in which these enzymes have the promiscuous activity that, when active, enable/increase growth on the specific nutrients: page 4. "Specifically, for four out of the five different substrate conditions, key mutations were linked to the predicted enzyme with promiscuous activity, which would be highly unlikely by chance ($P < 10^{-8}$, Fisher's exact test)."

Given these statistically sound results, we think it's highly unlikely that our model predictions are the result of a coincidental match between experiment and modeling.

5: I apologize if the critique regarding the contingency table in Fig S2 was imprecise. The main issue is that FucI, FucK and FucA genes were not observed in this work. They relate to a very old, previous observation (LeBlanc & Mortlock, 1971). As presented, it is now included as a

result in the statistical analysis. The authors need to be clear. Does the contingency table capture their own data? Or their own data and a thorough meta-analysis of published literature? If the second, a more extensive description of this meta-analysis and the associated statistical problems should be given. Alternatively, they should exclude FucI, FucK and FucA from their statistical analysis and discuss it separately as a previously published finding in the text.

We have made adjustments to the main text (bottom of page 4, top of page 5) and the Appendix Figure S2 caption to clarify the analysis. The observed mutations in the D-arabinose experiments in this study were in the fucR gene, a known DNA-binding transcriptional activator associated with

regulating the expression of the *fucAO* and *fucPIK* operons (Podolny et al. 1999). It was thus inferred that the strains in the D-arabinose evolution experiments were utilizing the *FucI*, *FucK*, *FucA* pathway to metabolize D-arabinose, in agreement with the supporting 1971 study. This is the inference which was made, which we are confident in linking given the clear results of the LeBlanc and Mortlock study.

We show further experimental evidence of this in our study, later on in the manuscript, in the analysis of *araC* mutations and in Figure 3D. Here we showed that our strains could not grow without the *fucK* gene; thereby, providing further support that these evolved strains were indeed utilizing the *fuc* operon genes for growth on D-arabinose. Thus, we disagree that the appendix figure should not include the *fucIKA* genes. Further, we would like to clarify to the reviewer that this contingency table is based on the experimental evidence of this study, as well as the computational predictions based on thorough database and literature mining, provided in the Notebaart et al. 2014 publication. Please see the above response to issue 2 for further details on the significance of the fisher's exact test contingency table results.

6 and 7: I'm not convinced by the 2-deoxyribose results. As stands the 2-deoxyribose results are entirely based on a single experimental population and clones with reconstructed presumed driver mutation did not grow on 2-deoxyribose. There is no buffer whatsoever against error - mislabelling, mix-up, contamination. Reproducibility is not a luxury item required for proper statistics, it is essential for science. I firmly believe that 2-deoxyribose results should either be reproduced at least once or should be removed from the paper. Moreover, as the 2-deoxyribose results are shaky and the presumed driver not confirmed, an extensive discussion of this mutation and its relation to selection seems premature.

We have conducted a follow-up experiment to specifically address the concern of a potential mislabelling, mix-up, contamination by reverse engineering the observed mutations in a clean ancestral strain. Please see Appendix Figure S4 and text that has been added to the last paragraph of the manuscript section "Modeling with underground metabolism accurately predicted key genes mutated during laboratory evolution experiments". Please see the highlighted text. To address the issues brought up by Reviewer 1, we grew a pORTMAGE library containing strains with the *RbsK* N20Y and *rbsR* insertion mutations separately and in combination. This library was grown alongside the wild type MG1655 ancestral strain on M9 minimal medium + 2 g/L D-2-deoxyribose agar plates. After incubation, sizable colonies were

visible on the pORTMAGE library half of the plates and no colonies were observed on the wild type side (Appendix Figure S4A). We selected 16 colonies for colony PCR and Sanger Sequencing and saw that all colonies contained both the *rbsR* insertion mutation and *RbsK* N20Y mutation, in addition to a third *rbsK* mutation (Appendix Figure S4B,C) similar to a duplication event observed in the original clone. Please see the text and Appendix Figure for further details. Overall, these results provide further evidence that the observed mutations in *rbsK* and *rbsR* enabled growth on the non-native D-2-deoxyribose substrate and that there was a strong selection pressure on the ribokinase underground activity. Further, there were multiple ways to impact *rbsK* as both duplication events and structural mutations (Table 1) or multiple structural mutations were separately observed in strains which grew on D-2-deoxyribose.

Accepted

14th March 2019

Thank you again for sending us your revised manuscript and the additional files. We are now satisfied with the modifications made and I am pleased to inform you that your paper has been accepted for publication.

Corresponding Author Name: Adam M Feist

Manuscript Number: MSB-18-8462